



# Development and evaluation of a new compact mechanism for aromatic oxidation in atmospheric models

Kelvin H. Bates[1,2], Daniel J. Jacob[1], Ke Li[3], Peter D. Ivatt[4,5], Mat J. Evans[4,5], Yingying Yan[6], and Jintai Lin[7]

[1]School of Engineering and Applied Sciences, Harvard University, Cambridge, MA, USA
[2]Department of Environmental Toxicology, University of California at Davis, Davis, CA, USA
[3]School of Environmental Science and Engineering, Nanjing University of Information Science and Technology, Nanjing, China
[4]Wolfson Atmospheric Chemistry Laboratories, Department of Chemistry, University of York, York, UK
[5]National Centre for Atmospheric Science, Department of Chemistry, University of York, York, UK
[6]Department of Atmospheric Sciences, School of Environmental Studies, China University of Geosciences (Wuhan), Wuhan, China
[7]Laboratory for Climate and Ocean–Atmosphere Studies, Department of Atmospheric and Oceanic Sciences, School of Physics, Peking University, Beijing, China

**Correspondence:** Kelvin H. Bates (kelvin_bates@fas.harvard.edu)

**Abstract.**

Aromatic hydrocarbons (mainly benzene, toluene, and xylenes) play an important role in atmospheric chemistry but the associated chemical mechanisms are complex and uncertain. Spare representation of this chemistry in models is needed for computational tractability. Here we develop a new compact mechanism for aromatic chemistry (GC13) that captures current

knowledge from laboratory and computational studies with only 17 unique species and 44 reactions. We compare GC13 to six other currently used mechanisms of varying complexity in box model simulations of environmental chamber data and diurnal boundary layer chemistry, and show that GC13 provides results consistent with or better than more complex mechanisms for oxygenated products (alcohols, carbonyls, dicarbonyls), ozone, and hydrogen oxide ($HO_x \equiv OH + HO_2$) radicals. GC13 features in particular increased radical recycling and increased ozone destruction from phenoxy-phenylperoxy radical cycling

relative to other mechanisms. We implement GC13 into the GEOS-Chem global chemical transport model and find higher glyoxal yields and net ozone loss from aromatic chemistry compared to other mechanisms. Aromatic oxidation in the model contributes 23%, 5%, and 8% of global glyoxal, methylglyoxal, and formic acid production respectively, and has mixed effects on formaldehyde. It drives small decreases in global tropospheric OH (-2.2%), $NO_x$ ($\equiv NO + NO_2$; -3.7%) and ozone (-0.8%), but a large increase in $NO_3$ (+22%) from phenoxy-phenylperoxy radical cycling. Regional effects in polluted environments can

be substantially larger, especially from photolysis of carbonyls produced by aromatic oxidation, which drives large wintertime increases in OH and ozone concentrations.





# 1 Introduction

Aromatic hydrocarbons are a major class of volatile organic compounds (VOCs) emitted to the atmosphere, with important effects on oxidant chemistry and secondary organic aerosol (SOA) formation. They are emitted from anthropogenic sources, including incomplete combustion, industrial processes, solvent evaporation, and fuel leakage (Na et al., 2004; Reimann and
Lewis, 2007), and have additional sources from biomass burning and vegetation (Misztal et al., 2015; Cabrera-Perez et al., 2016). Aromatics compounds in the gas phase can be directly harmful to human health (Duarte-Davidson et al., 2001; Sarigiannis and Gotti, 2008; Manuela et al., 2012) and their secondary chemistry can play a dominant role in ozone and SOA production in urban air (Barletta et al., 2005; Tan et al., 2012; Khan et al., 2018; Nault et al., 2018; Oak et al., 2019; Schroeder et al., 2020).

The simplest emitted aromatic hydrocarbons – benzene ($C_6H_6$), toluene ($C_7H_8$), and xylenes ($C_8H_{10}$), referred to collectively as BTX – together make up over 20% of global anthropogenic non-methane VOC emissions on a carbon basis (Yan et al., 2019) and up to 60% of urban emissions (Lee et al., 2002; Ran et al., 2009). The oxidative chemistry of BTX, initiated by reaction with the hydroxyl radical (OH), is unique among VOCs due to the stability of the aromatic ring and to their low H/C ratios. Standard chemical mechanisms for the fates of the peroxy radicals ($RO_2$) and alkoxy radicals (RO) produced in VOC
oxidation may thus not apply to aromatics (Vereecken, 2018; Xu et al., 2020). Low-volatility oxygenated organics produced from aromatic oxidation contribute to SOA formation (Ng et al., 2007; Hildebrandt et al., 2009; Schwantes et al., 2017) and may lead to new particle formation (Wang et al., 2017; Molteni et al., 2018; Garmash et al., 2020). Efficient production of peroxyacylnitrates (PANs) from BTX oxidation provides a reservoir of nitrogen oxide radicals ($NO_x \equiv NO + NO_2$), increasing ozone and OH on a global scale (Fischer et al., 2014). Interest in using satellite observations of formaldehyde ($CH_2O$) and gly-
oxal ($C_2H_2O_2$) as proxies of VOC emissions has further motivated the need to quantify yields of these species from aromatics (Chance et al., 2000; Wittrock et al., 2006; Liu et al., 2012; Chan Miller et al., 2016). Aromatics are particularly important for glyoxal and methylglyoxal production, providing another avenue for SOA formation (Fu et al., 2008; Lin et al., 2012).

The importance of aromatics for ozone formation spurred the initial development of BTX oxidation mechanisms for air quality models (Carter, 1990; Stockwell et al., 1997). The Master Chemical Mechanism (MCM) gives a quasi-explicit repre-
sentation of BTX atmospheric chemistry with thousands of reactions (Jenkin et al., 2003) but is computationally intractable for 3-D models. A range of simplified mechanisms are presently used in models (Jenkin et al., 2008; Goliff et al., 2013; Carter and Heo, 2013; Emmons et al., 2020) but can differ greatly in their results. Two recent studies implementing BTX chemistry into global models found opposite effects on global tropospheric ozone (Yan et al., 2019; Taraborrelli et al., 2021). Differences between mechanisms reflect evolution of knowledge as well as remaining uncertainties and parameterization choices
(Schwantes et al., 2017; Xu et al., 2020). Because of the high computational cost of chemical evolution and transport in 3-D models (Nielsen et al., 2017; Hu et al., 2018), it is imperative to minimize the number of species uniquely needed to describe aromatic chemistry (Stockwell et al., 2012; Brown-Steiner et al., 2018; Shen et al., 2020).

Here we present a new compact mechanism for BTX oxidation, GEOS-Chem version 13 (GC13), that is sufficiently simple for use in 3-D models but retains the accuracy of far more complex mechanisms and successfully fits laboratory data for BTX





oxidation products. GC13 incorporates new knowledge on phenoxy-phenylperoxy radical cycling (Taraborrelli et al., 2021), later-generation chemistry of hydroxylated aromatics (Schwantes et al., 2017) and fragmentation products (Newland et al., 2019; Wang et al., 2020), and increased radical cycling in the reactions of first-generation aromatic peroxy radicals (Xu et al., 2020). We evaluate GC13 in box model simulations of laboratory chamber experiments and the continental boundary layer and

compare it to six other mechanisms used in atmospheric models (CRI, MCM, MECCA, MOZART, RACM2, SAPRC). We implement GC13 into GEOS-Chem, a global chemical transport model (CTM), to diagnose the effects of aromatic chemistry in the troposphere on oxygenated organics and oxidant chemistry.

## 2   GC13: A new compact aromatic mechanism for atmospheric models

GC13 includes 17 unique species and 44 unique reactions to describe BTX chemistry. These are listed in Section S1 of the

Supplemental Information. Figures 1-3 show the dominant routes of BTX oxidation. Initial branching ratios in GC13 are shown in red, and species treated explicitly in blue. Starting from current knowledge on reaction pathways, we reduce the mechanism to be as simple as possible for atmospheric modeling while accurately representing important outcomes, including (a) ozone formation, (b) yields of major first-generation products, (c) short- and long-term yields of formaldehyde, glyoxal, and methylglyoxal, (d) effects on hydrogen oxide ($HO_x \equiv OH + HO_2$) radical budgets, and (e) closure of the total carbon budget.

While we do not yet represent SOA formation with GC13, we include intermediates and pathways by which SOA formation is known to occur, facilitating future implementation of an SOA module. We find that separately representing individual xylene isomers provides negligible benefits, and therefore represent them as a single lumped species. We also combing many xylene oxidation products with toluene products, scaling product yields to retain mass balance.

The first steps of BTX oxidation are summarized in Figure 1. OH is the only significant oxidant; lifetimes with respect to

oxidation at 298 K and $[OH] = 1 \times 10^6$ molecules cm$^{-3}$ are 9.6 d for benzene, 2.1 d for toluene, and 16 h for xylene (Mellouki et al., 2020). Reactions of BTX with $NO_3$ radicals are at least a factor of $10^5$ slower than their reactions with OH (Atkinson et al., 1984).

BTX oxidation by OH can proceed via either of two pathways: hydrogen abstraction from an alkyl substituent (**i**, only available to toluene and xylene), or OH addition to the aromatic ring (**ii**). Route **i** leads eventually to benzaldehyde (from

toluene) or tolualdehyde (from xylene) (**1a**); as described in Section 4.1, we find that experimental benzaldehyde yields are best fit by skipping the intermediate peroxy radical and proceeding directly to aldehyde formation. Route **ii** can either be followed by H-abstraction (**iii**), leading to a stable hydroxylated compound (**1b**, e.g. phenol), or by $O_2$ addition (**iv**), leading to the formation of a bridged bicyclic peroxy radical (**1c**). (For a more detailed description of the dynamic system of reversible $O_2$ addition, see Xu et al. (2020)). While other mechanisms and past studies have suggested the intermediate formation of

other products (pathway **v**) preceding **1c**, such as $C_6$ epoxides, these remain speculative or observed only under high-NO or low-pressure conditions unrepresentative of ambient BTX oxidation (Yu and Jeffries, 1997; Berndt and Böge, 2001; Birdsall and Elrod, 2011). Recent experimental and theoretical evidence suggests they do not form under ambient conditions (Wang et al., 2013; Zaytsev et al., 2019; Xu et al., 2020), so we assume that **1c** is the only product from route **iv**.




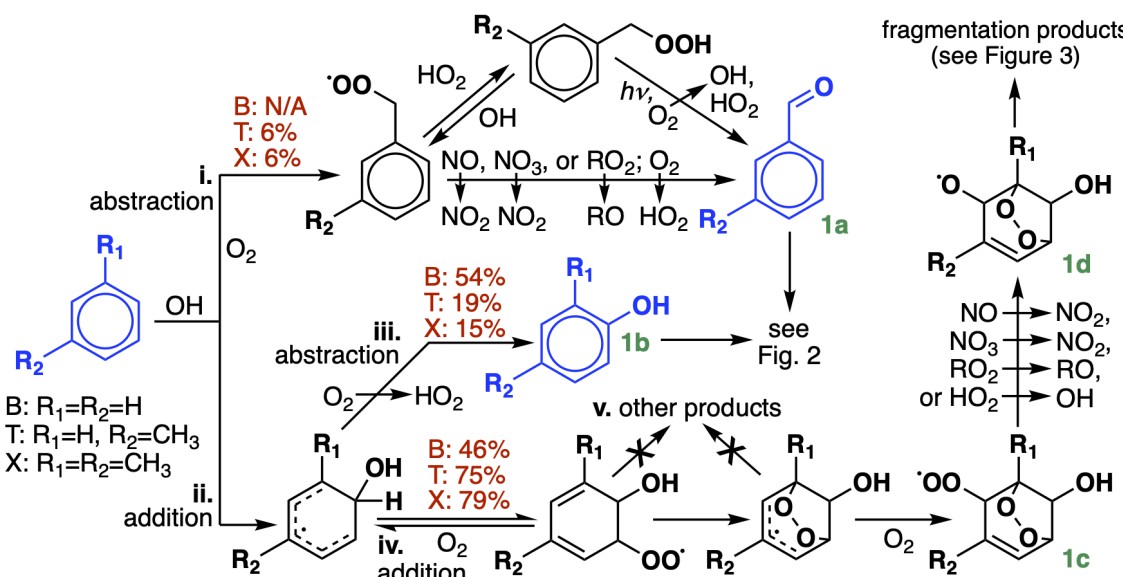

**Figure 1.** Initial oxidation chemistry of benzene, toluene, and xylenes. $R_1$ and $R_2$ denote either H or $CH_3$ depending on the species. For simplicity, only *m*-xylene is shown; in GC13, the xylene isomers are lumped together. Initial branching ratios in GC13 are shown in red, and species treated explicitly and uniquely in the aromatic oxidation mechanism are highlighted in blue.

The bridged bicyclic peroxy radical **1c** would traditionally be treated as a mechanistic branching point, leading potentially to the formation of a hydroperoxide, an organonitrate, an alcohol, or an alkoxy radical depending on the bimolecular reaction partner. However, recent work by Xu et al. (2020) found that nitrate and hydroperoxide yields from **1c** never exceeded 0.06% and 1% respectively for benzene. Instead, they suggest that reaction with either NO or $HO_2$ leads exclusively to alkoxy radical

formation (**1d**, with $NO_2$ and OH as respective coproducts) and subsequent fragmentation, and that this pattern should hold for larger aromatics as well. These higher-than-expected radical recycling rates are consistent with other recent results and hypotheses (Orlando and Tyndall, 2012; Praske et al., 2015; Zaytsev et al., 2019) and help to bring OH radical concentrations in line with values inferred from BTX decay rates in chamber experiments (Bloss et al., 2005a). In our mechanism, this means that we can bypass representation of the individual peroxy radical species and proceed straight to ring-opening products, using

a single generic peroxy radical species to accomplish conversion of bimolecular reaction partners (e.g., NO to $NO_2$).

The secondary chemistry of the ring-retaining aldehyde (**1a**) and alcohol (**1b**) products is shown in Figure 2. We lump benzaldehyde with tolualdehyde, given their similar chemistries. Benzaldehyde oxidation by either OH or $NO_3$ generates a peroxyacyl radical (**2a**) that can react with $NO_2$ to form peroxybenzoyl nitrate (**2b**), a PAN-like species we represent explicitly for its ability to sequester and transport $NO_x$. Reaction of **2a** with $HO_2$ yields perbenzoic acid (**2c**), benzoic acid and ozone,

or a benzoyloxy radical and OH, in a 65:15:20 ratio (Roth et al., 2010); subsequent chemistry leads eventually to formation of the phenylperoxy radical (**2d**). Benzaldehyde photolysis is also implemented with recently reevaluated rates (Mellouki et al., 2020), producing OH, $CO_2$, and the phenylperoxy radical (**2d**).




**Figure 2.** Oxidation chemistry of stable ring-retaining aldehydes and alcohols from benzene, toluene, and xylene. For simplicity, only benzaldehyde and phenol are shown; cresol chemistry is not shown, but is similar to phenol. GC13 lumps tolualdehyde with benzaldehyde, xylenols with cresol, and methylcatechols with catechol. Species treated explicitly and uniquely in the GC13 mechanism are highlighted in blue. The reaction of catechols (**2g**) with OH produces substituted phenoxy radicals, lumped here with the phenoxy radical (**2e**).

Rates and branching ratios for phenol and cresol oxidation by OH and $NO_3$ are taken directly from MCM. We lump xylenols with cresols, given their similar reactivity, and lump all products of cresol/xylenol oxidation with those of phenol. Both OH and $NO_3$ can abstract hydrogens, leading either to bridged peroxy radicals (**1c**) or to phenoxy radicals (**2e**), or undergo addition. $NO_3$ addition leads to ring-retaining organonitrates, which we lump with nitrophenols (**2f**), while OH addition forms catechols

(**2g**). While other mechanisms including MCM typically assume that the reaction of catechols with OH proceeds primarily by abstraction from the hydroxyl group, leading to functionalized phenoxy radicals, Schwantes et al. (2017) found that addition pathways dominate, leading to a proliferation of oxidized species that may act as SOA precursors. We cannot represent these species in detail in a condensed mechanism; future work will focus on parameterizing SOA yields from these pathways. Instead, we adopt branching ratios from Schwantes et al. (2017), lump polyhydroxylated aromatics with cresols and other

oxidized products with analogs, and adjust the cresol + OH reaction rate such that it represents only the abstraction pathways. H-abstraction from cresols would make functionalized phenoxy radicals, which we combine with the phenoxy radical in GC13. We also include the reactions of catechols with $NO_3$ and $O_3$, with rates from MCM and products lumped in with those from direct phenol and cresol oxidation.

      Both benzaldehyde and phenol/cresol oxidation lead eventually to the formation of phenylperoxy and/or phenoxy radicals

(**2d** & **2e**), the chemistry of which can have important implications for radical and ozone budgets via their cycling and formation of nitrophenols (**2f**; see Section 4.3). Due to its aromatic stability, the phenoxy radical is far more stable than a typical alkoxy radical, and rather than fragmenting, tends to react either with $NO_2$ (forming nitrophenols) or ozone (forming the phenylperoxy radical). The phenoxy and phenylperoxy radicals thus cycle until interrupted by phenyl hydroperoxide (**2h**) formation from phenylperoxy + $HO_2$, which is only a temporary radical reservoir, or by nitrophenol formation from phenoxy +





**Figure 3.** Chemical pathways following OH addition to BTX, continued from Figure 1. $R_1$ and $R_2$ denote either H or $CH_3$ depending on the species. For toluene and xylene, only OH addition ortho- to $R_1$ is shown, and for xylene, only the meta isomer is shown; isomeric differences could lead to differing placement of $R_1$ and $R_2$ on the stable products, including formation of acetic rather than formic acid (bottom right). Compounds in blue (**3c** and **3e**) are treated as lumped stable intermediates in GC13.

$NO_2$, which provides a terminal sink and the only route to fragmentation products from this radical system. Here, we treat the phenoxy-phenylperoxy system similarly to their representation in recent work by Taraborrelli et al. (2021), including the phenylperoxy + $NO_2$ reaction (Jagiella and Zabel, 2007), the phenoxy + $O_3$ reaction (Tao and Li, 1999), explicit treatment of phenyl hydroperoxide and nitrophenols (lumped with nitrocatechols), and HONO formation from nitrophenol photolysis
(Bejan et al., 2006; Chen et al., 2011).

   Finally, Figure 3 shows the chemical pathways of the bridged alkoxy radical **1d** produced following OH addition to BTX (Figure 1). These pathways remain uncertain, particularly for toluene and xylene, but represent the main source of glyoxal and methylglyoxal from BTX oxidation. A theoretical study by Wang et al. (2013) suggested ring-closure (**vi**) as the dominant fate of the bicyclic alkoxy radical from benzene, forming an epoxide (**3a**). However, follow-up calculations suggested instead that
ring-breaking (**vii**) dominates (98%+; Wang et al. (2020)), resulting in an acyclic alkoxy radical (**3b**), which was confirmed experimentally by Xu et al. (2020). The acyclic alkoxy radical can then decompose directly, forming a conjugated γ-dicarbonyl (**3c**) and glyoxal or methylglyoxal. Conjugated γ-dicarbonyls are highly photolabile, with typical daytime lifetimes of 10-15 minutes (Newland et al., 2019); however, their photolysis does not result in substantial $HO_x$ production, instead leading primarily to ketene-enol formation (**3e**), with a minor route decomposing to CO and an acrolein derivative (Newland et al.,
2019).

   Recent theoretical (Wang et al., 2020) and experimental (Xu et al., 2020) studies have shown that instead of directly decomposing, the acyclic alkoxy radical (**3b**) can undergo a 1,5 aldehydic H-shift (**viii**), yielding a resonance-stabilized allyl radical (**3d**). This allyl radical represents a mechanistic branching point with many possible fates, dominated by decomposition to an α-dicarbonyl and a ketene-enol (**3e**). Between this direct production and secondary formation from γ-dicarbonyl photolysis,





Wang et al. (2020) and Xu et al. (2020) suggest that ketene-enols are the primary product of the ring-opening pathway; however, Xu et al. (2020) note that they do not observe unity yields from benzene, and suggest that other reactive pathways from **3d** may account for the missing carbon. Such pathways include CO elimination and $O_2$ addition, the latter of which could lead to autoxidation and the formation of highly oxidized SOA precursors from aromatics (Wang et al., 2017; Molteni et al., 2018; Garmash et al., 2020).

Subsequent chemistry of the ketene-enols (**3e**) was studied by Newland et al. (2019) and Wang et al. (2020), and includes unimolecular cyclization to furanones as well as direct formation of formic and acetic acids via reaction with OH and $O_3$. Newland et al. (2019) also observed tautomerization of the ketene-enols to ketene-carbonyls, possibly mediated by chamber surfaces, followed by cyclization to form anhydrides. Further studies are needed to constrain the contributions of these pathways, particularly for ring-opening products from toluene and xylene.

Because the branching ratios of the ring-opening pathways in Figure 3 remain poorly constrained, particularly for toluene and xylene, we do not speciate them in detail in GC13. Instead, we tune the yields of glyoxal and methylglyoxal (coproduced with **3c** and **3e**) to match observed yields from chamber experiments, and lump larger products into two representative intermediates, which stand in for the $C_{4+}$ stable products in Figure 3 (primarily **3c** and **3e**). One, treated as a $C_4$ compound and produced from all three BTX precursors, yields glyoxal and other compounds lacking methyl groups upon its subsequent oxidation, while the other, treated as a $C_5$ compound and produced only from toluene and xylene, also yields methylglyoxal and other methylated products. The subsequent chemistry of these representative $C_4$-$C_5$ intermediates is a weighted combination of the reactive fates of the conjugated dicarbonyls (**3c**) and the ketene-enols (**3e**), drawn from Newland et al. (2019) and MCM for **3c** and from Newland et al. (2019) and Wang et al. (2020) for **3e**, adjusted slightly to tune the later-generation yields of glyoxal and methylglyoxal. We do not explicitly track the furanones and anhydrides produced in unimolecular rearrangements of the ketene-enols (**ix**), allocating this carbon instead to other oxygenated intermediates with similar functionalities and lifetimes while maintaining carbon balance. We include a route to direct CO formation from **3d**, using a branching ratio from Xu et al. (2020) for benzene, but do not include the minor pathway to highly oxidized molecule (HOM) formation; subsequent updates to the mechanism focused on SOA formation could represent these pathways explicitly.

## 3   Previous aromatic mechanisms

In the following sections, we will compare GC13 to a suite of commonly used aromatic mechanisms. Here, we briefly describe each mechanism, in order from most to least complex. The number of species and reactions in each mechanism, excluding inorganic species and generic $C_1$-$C_3$ compounds, are given in Table 1. The mechanisms span a wide range of complexity, with two orders of magnitude separating the numbers of species and reactions in the largest and smallest mechanisms.

*MCM*. The Master Chemical Mechanism (MCM) is a near-explicit mechanism that treats the full oxidative degradation of benzene, toluene, and each xylene isomer. The mechanism was developed by Jenkin et al. (2003) and updated to version 3.1 in Bloss et al. (2005b) based on an assessment of experimental work. Bloss et al. (2005a, b) compared the mechanism to a





**Table 1.** Sizes of the aromatic schemes in common mechanisms[a]

| Mechanism | Number of: | | Reference |
| | Species | Reactions | |
| --- | --- | --- | --- |
| MCM v3.1 | 1271 | 3788 | Bloss et al. (2005b) |
| MECCA | 229 | 666 | Taraborrelli et al. (2021) |
| SAPRC-11 | 55 | 374 | Carter and Heo (2013) |
| CRI v2-R5 | 56 | 128 | Watson et al. (2008) |
| RACM2 | 34 | 115 | Goliff et al. (2013) |
| MOZART-T1 | 33 | 56 | Emmons et al. (2020) |
| MOZART-GC | 13 | 43 | Porter et al. (2017) |
| GC13 | 17 | 44 | This work |

[a] Mechanisms used in intercomparison to GC13. Species count includes both stable and
radical species, but does not include inorganic reactants and common/generic $C_1$-$C_4$
species. Reaction count does not include reactions that only contain these excluded
species.

series of chamber experiments and noted that, while ozone was well-simulated in benzene oxidation, simulations typically
overestimated ozone formation from the larger aromatics while underestimating OH concentrations in all experiments.

*MECCA*. The Module Efficiently Calculating the Chemistry of the Atmosphere (MECCA; Sander et al. (2011)) includes
detailed aromatic chemistry described by Cabrera-Perez et al. (2016). The chemistry of toluene and benzene is taken from
MCM without simplification, but with some important updates (notably to phenoxy-phenylperoxy radical cycling) described in
detail by Cabrera-Perez et al. (2016) and Taraborrelli et al. (2021). MECCA lumps the xylene isomers and sets their chemistry
identical to that of toluene, aside from the branching ratios in their initial reaction with OH. Despite its large number of species
and reactions, MECCA has been used in a global 3-D model study by Taraborrelli et al. (2021).

*SAPRC*. The Statewide Air Pollution Research Center (SAPRC) mechanisms are a family of moderately reduced mecha-
nisms widely used in airshed models for the prediction of ozone formation and representation of organic pollutants. Here we
use SAPRC-11 (Carter and Heo, 2013), which was specifically designed to optimize aromatic oxidation and ozone forma-
tion rates in comparison to environmental chamber experiments. SAPRC-11 represents the xylene isomers individually and
includes many early-generation BTX oxidation products, including phenol, cresol, xylenols, catechol, dicarbonyl compounds,
nitrophenols, and benzaldehyde. SAPRC-11 was previously implemented in GEOS-Chem (v9-02) by Yan et al. (2019). Ver-
sions of SAPRC have also been implemented in the Weather Research and Forecasting chemistry model (WRF-Chem), the
Community Multiscale Air Quality (CMAQ) model, and other regional 3D models (Yu et al., 2010; Cai et al., 2011; Zhang
et al., 2012; Kitayama et al., 2019; Shareef et al., 2019).

*CRI*. The Common Representative Intermediates (CRI) mechanism, first developed by Jenkin et al. (2002), is a reduced
mechanism based on MCM that uses a series of generalized intermediates, rather than explicitly simulating multigenerational
chemistry, to simulate ozone formation rates. Optimization and comparisons to MCM for the most recent version, v2, are





described by Jenkin et al. (2008). Here, we use CRI v2-R5, a further reduction that lumps together the xylene isomers (Watson et al., 2008). CRI v2-R5 has been widely implemented in regional and global models, including WRF-Chem (Archer-Nicholls et al., 2014), STOCHEM (Utembe et al., 2010; Khan et al., 2015), and UKCA (Archer-Nicholls et al., 2020).

*RACM.* The Regional Atmospheric Chemistry Mechanism (RACM) is a reduced mechanism first presented by Stockwell
et al. (1997) and intended for use in modeling gas-phase chemistry in a wide range of ambient conditions. Here we use RACM2 as described by Goliff et al. (2013). RACM2 is based on MCM for benzene and on Calvert (2002) for toluene and the individual xylene isomers, and represents major first-generation products of BTX oxidation, including benzaldehyde, phenol, cresol, an epoxide, and photolabile dicarbonyls. RACM is implemented in both WRF-Chem and CMAQ (Sarwar et al., 2013; Kitayama et al., 2019; Shareef et al., 2019).

*MOZART.* The Model for Ozone and Related chemical Tracers (MOZART) chemical mechanisms are designed for implementation in the NCAR Community Earth System Model (CESM). The most recent iteration, MOZART-T1 (Emmons et al., 2020), is the first to differentiate between the BTX compounds. With an oxidation scheme based on MCM, it represents some major early-generation products explicitly, including benzaldehyde, photolabile dicarbonyls, and peroxybenzoyl nitrate, but ignores some later-generation products assumed to undergo efficient deposition or aerosol uptake. An earlier version of the
aromatic chemistry in MOZART-T1, first described by Knote et al. (2014), was implemented into GEOS-Chem by Porter et al. (2017); for the sake of comparison and contextualization of past studies, we also implement the Porter et al. (2017) mechanism here, which we label "MOZART-GC".

# 4 Mechanism evaluation and intercomparison

## 4.1 Methods

We implemented GC13 and the other mechanisms of Section 3 into box model simulations for comparisons to environmental chamber data and for mechanism intercomparisons under a range of boundary layer conditions. The simulations use a $4^{th}$-order Rosenbrock kinetic solver implemented with the Kinetic PreProcessor tool (KPP; Damian et al. (2002); Daescu et al. (2003); Sandu et al. (2003)). We standardize the inorganic and $C_1$-$C_3$ chemistry to that of MCM in all mechanisms so that the only differences between the mechanisms are in their BTX oxidation chemistry. For mechanisms with speciated xylenes, we assume
equal contributions from the three isomers, and comparisons of xylene product yields with literature values are only conducted for experimental studies that targeted all three isomers.

Environmental chamber simulations. To quantify product yields and compare them to environmental chamber data, we simulate BTX chemistry in a box model representative of chamber experimental conditions. For each mechanism, we initialize the box model with fixed mixing ratios of one aromatic precursor, NO, and $H_2O_2$ as a photolytic OH source, then run the
simulation forward in time with a fixed light intensity and temperature until the aromatic precursor is 99% depleted. We then vary all initial settings (temperature, light intensity, and each reactant concentration) individually and rerun the simulation to sample the full range of possible experimental conditions. Result shown in Sections 4.2-4.3 are for simulations with initial $[NO]_0$ = 5 ppt – 2.5 ppm, P = 1 atm, T = 298 K, $[VOC]_0$ = 100 ppb, $[H_2O_2]_0$ = 2.5 ppm, and an $NO_2$ photolysis rate ($j_{NO2}$)





of $8 \times 10^{-3}$ s$^{-1}$ (other photolysis rates, given in Section S1 of the SI, are scaled to $j_{NO2}$); "initial" yields are after 10 min of photooxidation. Sensitivities to temperature and initial VOC concentrations are generally small, but additional results showing the effects of these parameters can be found in the SI. Wall loss is not represented in these simulations, but is also expected to play a minor role in the short-term chemistry described here.

Continental boundary layer simulations. To examine longer-term product yields and effects of BTX oxidation on the ambient atmosphere, the same box model described above is also run under conditions meant to simulate a continental boundary layer like that of the heavily studied Seoul Metropolitan Area (Oak et al., 2019; Schroeder et al., 2020) with constant NO and aromatic emissions. The well-mixed boundary layer exchanges with the background free troposphere with a fixed ventilation timescale of 1 d for all species. Simulations are initialized with 75 ppb O$_3$, 1.8 ppm CH$_4$, 200 ppb CO, 300 ppt CH$_2$O, and

1% H$_2$O, and these species are also present in the same concentrations in the background free troposphere with which the boundary layer box exchanges. Photolysis rates follow a clear-sky diurnal profile at 45° latitude at the summer solstice with an ozone column of 350 DU, while temperature varies sinusoidally with an amplitude of 4 °C, centered at 25 °C, peaking at 13:00 solar time, and a period of 1 d. Results shown in Sections 4.2-4.3 are for a total aromatic VOC emission rate of 120 ppt h$^{-1}$, distributed between benzene, toluene, and xylene in a 2:2:1 molar ratio or for a single precursor, and for fixed NO

emission rates between 1 ppt h$^{-1}$ and 10 ppb h$^{-1}$. Additional results showing sensitivities to VOC emission rates can be found in the SI. The model does not represent deposition or aerosol uptake processes, except to impose a 1 h loss rate on N$_2$O$_5$ for conversion to HNO$_3$. We apply 7 days of initialization to reach diurnal steady state; the results shown in Section 4.3 are from the 8$^{th}$ simulated day.

## 4.2  Oxygenated VOC yields

We use environmental chamber simulations to determine prompt product yields from each mechanism and compare them to experimental data, and use both chamber and boundary layer simulations to investigate the differences in long-term yields of later- and multi-generational products between mechanisms. In both cases we present our results as a function of NO$_x$ concentrations imposed in the model either directly (chamber simulations) or through emissions (boundary layer simulations). In some cases, yields from organic peroxy radical reactions may depend on the branching between bimolecular and unimolec-

ular reactions in addition to the branching between reaction with NO and other reactions, and comparisons would be more appropriately made as a function of peroxy radical lifetime against all bimolecular reactions rather than just NO (see, e.g., Xu et al. (2020)); however, it is difficult to approximate these bimolecular lifetime conditions for many past experimental results, so we opt for the more straightforward comparisons as a function of initial NO$_x$. In comparisons with experimental yields (Figures 4-6), simulated yields are shown after 20 min of oxidation, while experimental yields are the earliest reported value.

Experiments conducted with no added NO$_x$ are shown at [NO$_x$]$_0$ = 10 ppt, below which modeled yields are invariant with [NO$_x$]$_0$.

Ring-retaining products. Figure 4 shows environmental chamber molar yields of ring-retaining alcohols and aldehydes from BTX oxidation in the mechanisms and in the experimental literature as functions of initial NO$_x$ concentrations. For the ring-retaining alcohols formed via OH addition followed by H-abstraction (Figure 1**iii**), all mechanisms implement fixed direct





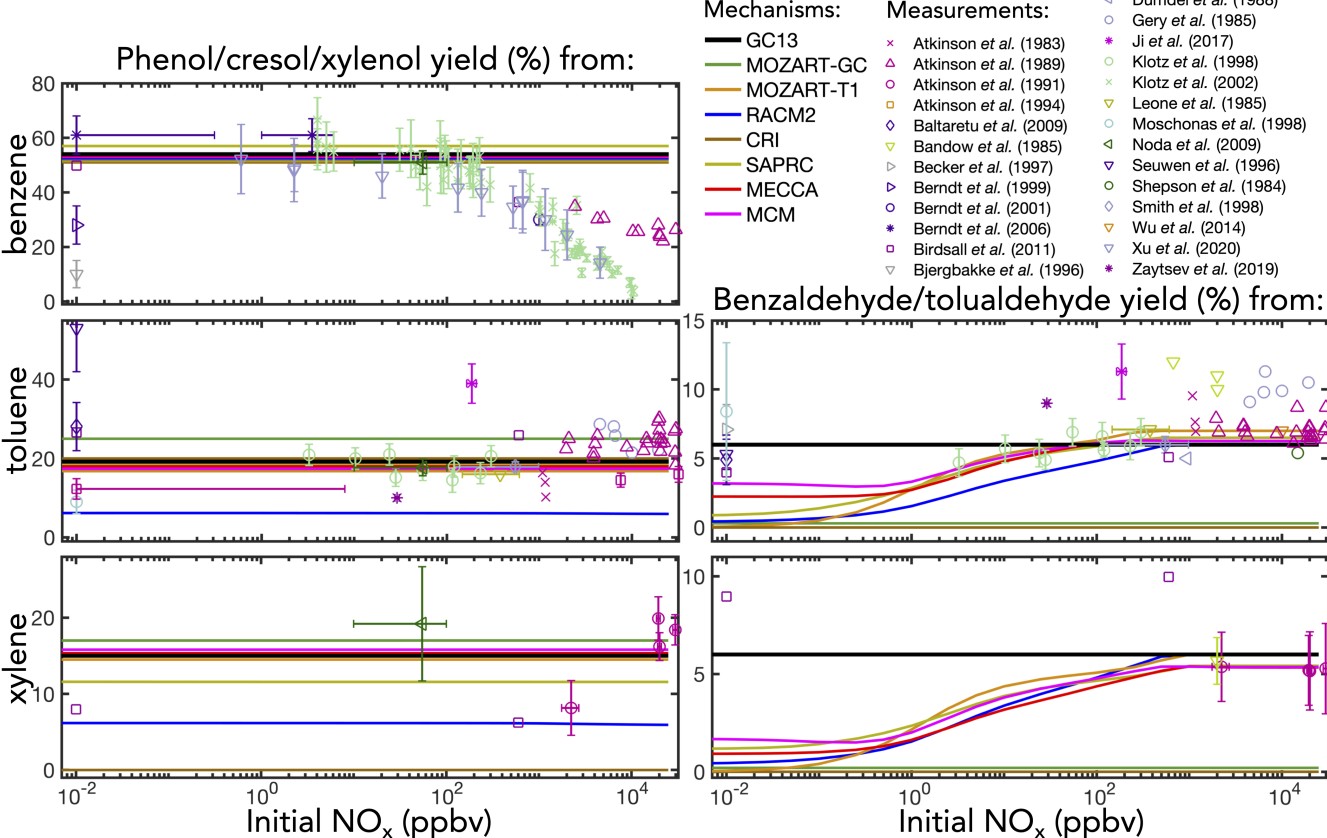

**Figure 4.** Experimental (points) and simulated (lines) prompt molar yields of aromatic alcohols and aldehydes from BTX oxidation in environmental chambers as a function of initial $NO_x$ mixing ratio. Overlapping simulation lines are offset slightly for visibility.

yields from the BTX precursors, which therefore do not vary with NO. GC13 uses a phenol yield from benzene of 54%, derived from an error-weighted average of the yields measured by Klotz et al. (2002) (53±7%), Berndt and Böge (2006) (61±7%), and Xu et al. (2020) (49±13%) at atmospherically relevant $NO_x$ levels. Among the other mechanisms implemented here, SAPRC uses a fixed phenol yield of 57% from benzene, while all others use 53%. Fixed cresol and xylenol yields of 19% and 15% in GC13, from toluene and xylene respectively, are taken from MECCA and are consistent with observations.

For phenol, the mechanisms' fixed yields of 53-57% fit most data under low-to-moderate NO conditions, but at high NO the observed yield declines as bimolecular reactions become too fast to allow equilibration to occur. This behavior is not exhibited by any of the mechanisms studied here; however, such high NO concentrations are rarely seen in ambient environments, and the fixed yields are therefore suitable for atmospheric simulations. Experimental yields of cresol (from toluene) and xylenol (from xylene) exhibit little correlation with $NO_x$, and are therefore adequately represented by fixed branching ratios, although the ratios implemented vary widely between mechanisms, with low yields in RACM2 (6% from toluene and xylene) and no representation of xylenols in CRI v2-R5.





The ring-retaining aldehydes benzaldehyde and tolualdehyde are formed following H-abstraction from the methyl groups of toluene and xylene respectively (Figure 1i). Most mechanisms that include benzaldehyde chemistry (CRI and MOZART-GC do not) explicitly treat the benzyl peroxy intermediate in this reaction pathway, resulting in peak aldehyde yields of ~6% at high $NO_x$ dropping to 1-3% at low $NO_x$ due to competing formation of benzyl hydroperoxide. However, this results in lower

benzaldehyde yields than observed under low-$NO_x$ conditions. The lack of observed $NO_x$-dependence in the benzaldehyde yield may reflect a short lifetime of benzyl hydroperoxide, a high incidence of $RO_2 + RO_2$ reactions (Moschonas et al., 1999), a non-hydroperoxide-forming channel in the benzyl peroxy + $HO_2$ reaction (Baltaretu et al., 2009), or another pathway altogether (Salta et al., 2020). Regardless, to fit observations and further simplify the mechanism, we bypass the benzyl peroxy intermediate and form benzaldehyde directly from the reaction of toluene and xylene with OH with a fixed yield of 6%,

consistent with most experimental results at both high and low $NO_x$.

$C_1$-$C_3$ *carbonyl products.* Figure 5 shows prompt environmental chamber yields of glyoxal from BTX oxidation in both the mechanisms and the experimental literature as functions of initial $NO_x$ concentrations (initialized as NO in simulations and in most chamber experiments, with some $NO_2$ in chamber experiments). Initial glyoxal yields from BTX oxidation generally range between 10% and 40%, with the highest yields from benzene and the lowest from xylenes. With a fixed $NO_x$-independent

first-generation glyoxal yield and secondary yields from the representative $C_4$-$C_5$ intermediates (Figure 3), GC13 accurately simulates experimentally measured glyoxal yields across a wide range of $[NO_x]_0$ for all three aromatic precursors, while most other mechanisms exhibit excessive glyoxal formation under high-$NO_x$ conditions and insufficient production under low-$NO_x$ conditions. Due to the long lifetime of benzene relative to its intermediate oxidation products, it is difficult to isolate prompt vs. multigenerational glyoxal yields, which explains the range of experimental results; the prompt yield we implement in GC13

(18%) is able to match the lower limit of observed yields at both high and low $NO_x$.

Figure 6 shows simulated and observed environmental chamber yields of formaldehyde and methylglyoxal from toluene and xylene oxidation as functions of initial $NO_x$ concentrations. Because it lacks alkyl substituents, benzene is unable to produce either formaldehyde or methylglyoxal. As with glyoxal, GC13 is consistent with experimental yields across the full spectrum of $[NO_x]_0$ conditions, although some observed methylglyoxal yields from toluene deviate considerably from the general trend.

Most other mechanisms exhibit lower prompt formaldehyde and methylglyoxal yields under low-$NO_x$ conditions and higher yields under high-$NO_x$ conditions than GC13 or observed values, although observational evidence is sparse for xylene. In particular, SAPRC and MCM exhibit high prompt formaldehyde and methylglyoxal yields, respectively, under high-$NO_x$ conditions.

In addition to these prompt yields, later-generation formation of $C_1$-$C_3$ carbonyl species can be important for $HO_x$ radical

and carbon budgets. Data on long-term yields are sparse, so here we rely mostly on model intercomparisons of boundary layer simulations with MCM taken as a reference. Results are shown in Figure 7 for mixed aromatic emissions, and in Figure S6 in the Supporting Information for individual aromatic precursors. The maxima in yields at intermediate $NO_x$, particularly evident for formaldehyde, reflect the corresponding maxima of OH concentrations. Generally, MCM is able to produce higher late-generation yields of the $C_1$-$C_3$ carbonyls than more reduced mechanisms, but GC13 exhibits high yields similar to MCM.

GC13 also simulates similar midday glyoxal-to-formaldehyde concentration ratios from aromatic oxidation to MCM, ranging





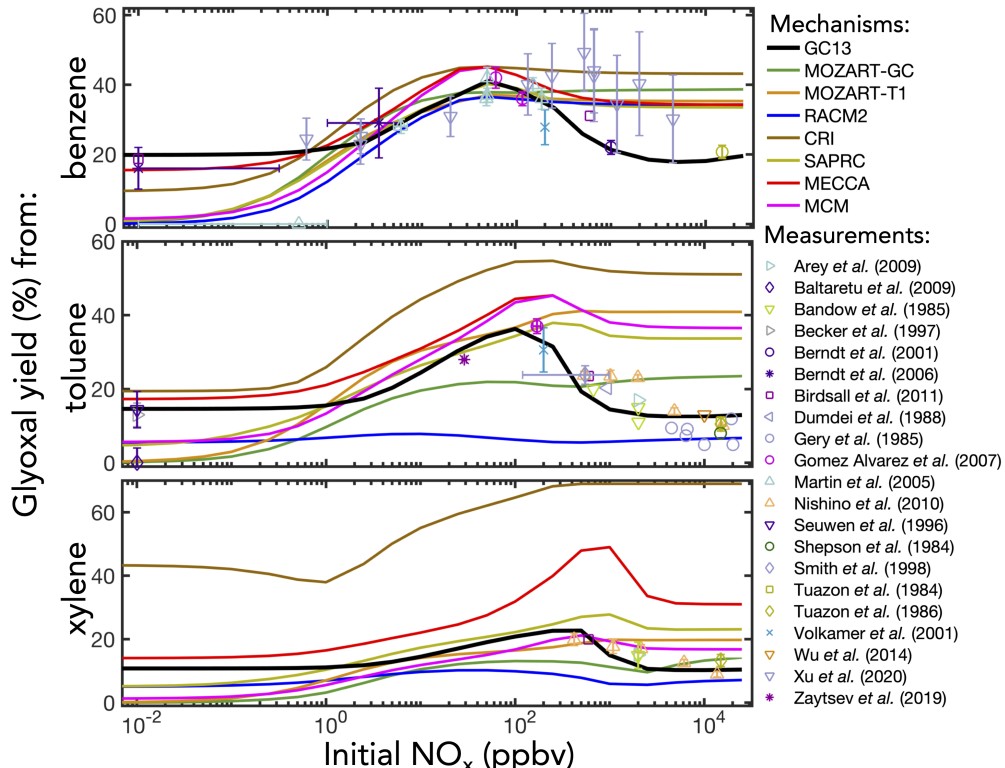

**Figure 5.** Experimental (points) and simulated (lines) prompt molar yields of glyoxal from BTX oxidation in environmental chambers as a function of initial $NO_x$ mixing ratio.

from 0.3 under high $NO_x$ conditions to >0.7 at low $NO_x$ (Figure S8). Results speciated by BTX precursor reveal a similar trend. MOZART-GC and MOZART-T1, the most reduced of previous mechanisms, tend to simulate the lowest long-term carbonyl yields. MECCA is generally able to reproduce MCM's high long-term carbonyl yields except from xylene, which MECCA lumps as toluene; this results in an overprediction of glyoxal yields and underprediction of formaldehyde and methylglyoxal

5    yields relative to MCM. Long-term yields from simulated chamber experiments (Figures S3-S4) show results similar to the boundary layer simulations.

*Other VOC products.* Some experimental and theoretical evidence exists for formation pathways of fumaraldehydic, formic, and acetic acids from BTX oxidation (Berndt et al., 1999; Dumdei and Kenny, 1988; Wang et al., 2020; Xu et al., 2020), but this is generally not included in mechanisms. Global models tend to underestimate ambient concentrations of these compounds, and

10    additional formation pathways could help alleviate this discrepancy (Millet et al., 2015; Khan et al., 2018). Here, we include the formation of formic and acetic acids as described in Newland et al. (2019) and Wang et al. (2020) via the ketene-enols, represented as part of the lumped $C_4$ and $C_5$ products. Few chamber data are available for comparison; Berndt et al. (1999) measured a 13% yield of formic acid from benzene under low-NO conditions, while Dumdei and Kenny (1988) measured a 6% yield of acetic acid from toluene under high-NO conditions. Both are consistent with prompt yields in GC13. Long-term



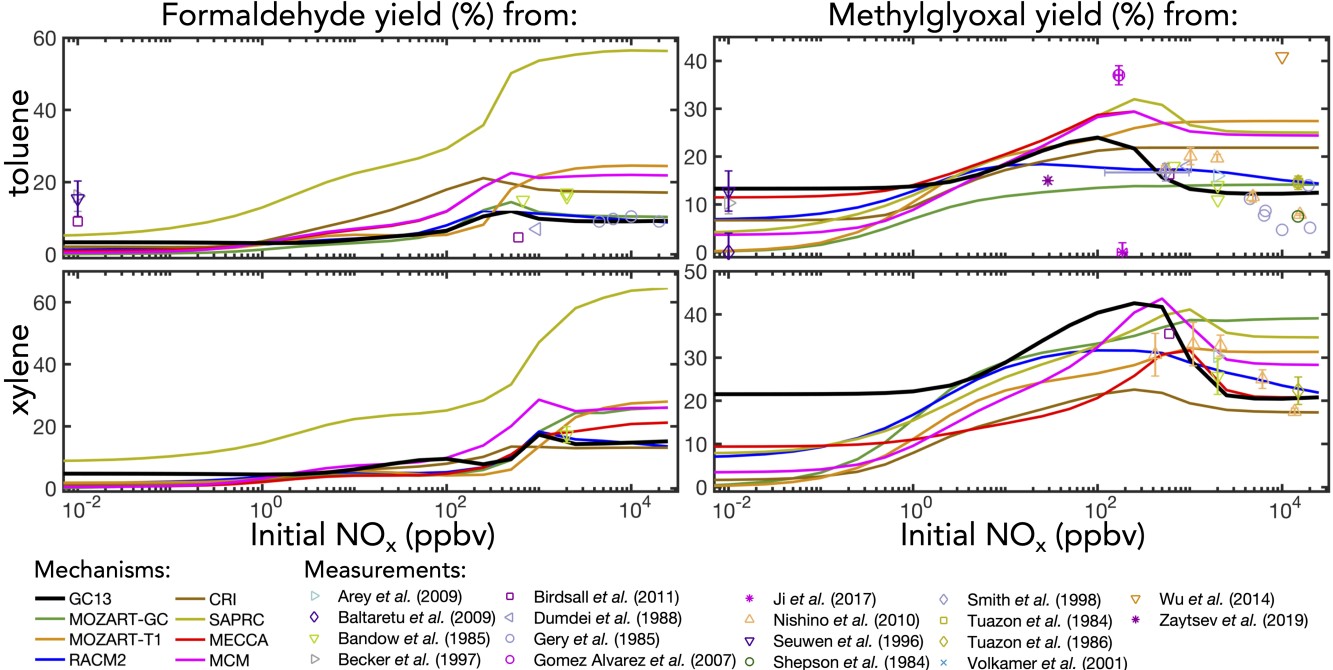

**Figure 6.** Experimental (points) and simulated (lines) prompt molar yields of methylglyoxal and formaldehyde from toluene and xylene oxidation in environmental chambers as a function of initial $NO_x$ mixing ratio.

yields of formic acid in the mechanism can reach 32% from benzene, 28% from toluene, and 16% from xylene, while acetic acid yields can reach 12-13% from toluene and xylene (See Figures S2 and S5).

Most mechanisms also represent the formation of larger ($C_{4+}$) dicarbonyls, including biacetyl, photolabile conjugated di-aldehydes, and less reactive ketones, with varying degrees of complexity. In GC13, these compounds are not treated explicitly, and are instead grouped into the two representative $C_4$-$C_5$ reactive intermediates. For this reason, we do not optimize any branching ratios in GC13 with environmental chamber yields of $C_{4+}$ dicarbonyls, but Figure S1 in the Supporting Information provides a comparison between measured and simulated yields of photolabile dicarbonyls in each mechanism.

Finally, as described in Section 2, most mechanisms include the direct formation of a ring-opened $C_{6+}$ epoxide from BTX oxidation. These yields are $NO_x$-independent and span a wide range, from 0% from all precursors in GC13 to >70% from xylene in RACM. While little evidence for this pathway exists under atmospherically relevant conditions, it may be useful as an intermediate, particularly given that other known pathways cannot achieve carbon closure in many experimental studies (e.g. Xu et al. (2020)). Although these epoxides are not included in GC13, Figure S1 provides a comparison between measured yields in environmental chambers and simulated yields in other mechanisms.





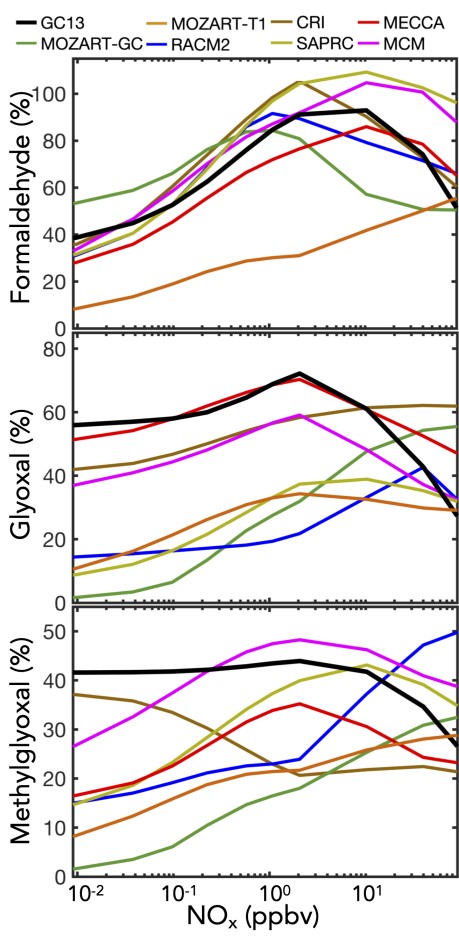

**Figure 7.** Simulated molar yields of formaldehyde, glyoxal, and methylglyoxal from BTX oxidation in a continental boundary layer as a function of midday $NO_x$ mixing ratio. Model setup is described in Section 4.1. Molar yields are averaged over the $8^{th}$ simulated day.

## 4.3 Effects on oxidants

Figure 8 shows the effects of BTX oxidation on daytime abundances of ozone, OH, and $HO_2$ in the continental boundary layer simulations. Results speciated by aromatic precursor are shown in Figure S7. GC13 and MECCA have less ozone production than MCM, RACM, and SAPRC, all of which are known to overestimate peak ozone from chamber experiments (Bloss et al.,

5   2005a; Goliff et al., 2013; Carter and Heo, 2013). This is primarily due to phenoxy-phenylperoxy radical cycling chemistry, represented only in GC13 and MECCA (Sander et al., 2019; Taraborrelli et al., 2021). The dominant phenoxy-phenylperoxy cycle converts $NO_2$ to $NO_3$ and consumes one ozone molecule; during the day, this cycle largely balances via $NO_3$ photolysis (although the minor channel to $NO + O_2$ represents an odd oxygen sink), but at night, conversion of $NO_3$ to $N_2O_5$ and on to $HNO_3$ amplifies ozone loss. MECCA and GC13 both exhibit a surge of $NO_3$ (and corresponding reduction in $NO_2$) at sunset,

10   which propagates to lower $NO_2$ and ozone levels throughout the following day (Figure S30). Flux through this phenoxy-





phenylperoxy system is highest from benzene because of the high phenol yields (Figure 1), so the ozone differences between mechanisms are strongest for benzene (see Figure S7).

GC13 also simulates higher $HO_x$ concentrations than other mechanisms, especially under low $NO_x$ conditions. This effect is due primarily to increased radical propagation from the bimolecular reactions of the bridged bicyclic peroxy radicals (Figure 1c), which do not form radical-terminating hydroperoxides or organonitrates in GC13. Higher radical recycling from subsequent reactions of the representative $C_4$ and $C_5$ intermediates also contributes. It has been reported that other mechanisms tend to underpredict $HO_x$ concentrations in simulations of chamber experiments (Chen, 2008), and both Bloss et al. (2005a) and Carter and Heo (2013) comment on the need to increase $HO_x$ recycling from aromatic oxidation; GC13 thus brings $HO_x$ concentrations into improved alignment with chamber results. These effects are stronger for toluene and xylene because of their higher bridged bicyclic peroxy radical and $C_4$-$C_5$ intermediate yields relative to benzene (see Figure S7).

To test the sensitivities of these outcomes to specific aspects of the GC13 mechanism, we conduct additional simulations with individually perturbed reaction rate constants and yields. Results from the sensitivity simulations with the most prominent changes are shown in Figure 9. We find that ozone is most sensitive to changes in the rates of the key reactions in the phenoxy-phenylperoxy system. These rates remain uncertain; Tao and Li (1999) measured a rate of 2.86 ($\pm 0.35$) $\times 10^{-13}$ cm$^3$ molecule$^{-1}$ s$^{-1}$ at 298 K for the $C_6H_5O + O_3$ reaction, which we use in GC13, but noted that it might be a lower limit, while Jagiella and Zabel (2007) did not set uncertainty bounds on their best fit rate constant of $7 \times 10^{-12}$ cm$^3$ molecule$^{-1}$ s$^{-1}$ for $C_6H_5O_2 + NO_2$ (used in GC13), instead only specifying a minimum of $1 \times 10^{-12}$ cm$^3$ molecule$^{-1}$ s$^{-1}$ consistent with their results. Increasing either rate by a factor of ten substantially increases ozone loss due to phenoxy-phenylperoxy cycling, highlighting the importance of better constraints on these rates.

Ozone and OH are both sensitive to changes in the fates of catechols and methylcatechols. While most mechanisms assume that the reactions of catechols and methylcatechols with OH proceed by abstraction to form functionalized phenoxy radicals, Schwantes et al. (2017) showed that addition pathways dominate, leading to heavily substituted low-volatility products that may contribute to SOA formation. In a sensitivity simulation with the product channels from catechols + OH turned off (representing complete loss of products to aerosols), the effects of aromatic oxidation on ozone production and OH are strongly diminished, as this represents a major loss of later-generation gas-phase products such as the $C_1$-$C_3$ carbonyls and $C_4$-$C_5$ intermediates. For both catechol and phenoxy-phenylperoxy perturbations, benzene is the most sensitive of the primary aromatics, due to its higher yields of the phenolic pathway than toluene or xylene (Figure S25).

$HO_x$ concentrations are also highly sensitive to the relative contributions of the radical propagation and termination pathways from the reactions of $HO_2$ with the initial bridged bicyclic peroxy radicals from BTX + OH under low-$NO_x$ conditions. A perturbation simulation with 100% radical termination (i.e. hydroperoxide formation), as in most mechanisms, reduces $HO_2$ in GC13 to levels similar to the other mechanisms, and causes a smaller reduction in OH, highlighting the importance of this branching ratio. However, Xu et al. (2020) showed that hydroperoxide formation is minimal from the benzene-derived bridged bicyclic peroxy radical, motivating our treatment in GC13.

Results from additional sensitivity simulations are shown in Section S5 of the Supporting Information (Figures S25-S29). Changes to other reactions in the phenoxy-phenylperoxy system can be important – perturbations to the $C_6H_5O + NO_2$ rate





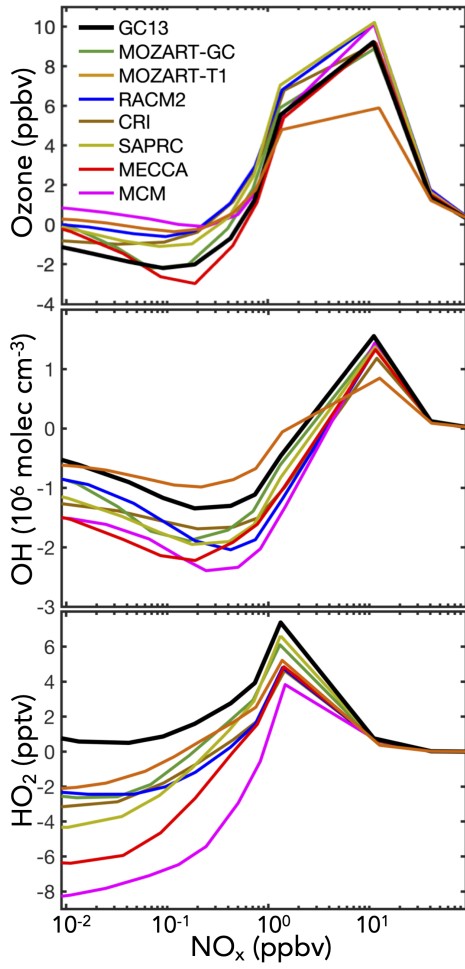

**Figure 8.** Changes in midday ozone, OH, and $HO_2$ concentrations due to aromatic chemistry in box model simulations of the continental boundary layer with different aromatic oxidation mechanisms. Changes in species concentrations are calculated by subtracting their midday (10:00-14:00) mean values in a simulation without aromatic emissions from those in an equivalent simulation with aromatic emissions, and are plotted against midday $NO_x$ mixing ratio. Aromatic emissions are 120 ppt h$^{-1}$ with 2:2:1 molar ratios for benzene:toluene:xylenes. Other model setup is described in Section 4.1.

have similar effects to those of $C_6H_5O + O_3$ because the two reactions are in direct competition, while increasing the $C_6H_5O_2$ + $NO_3$ rate increases ozone loss in a manner similar to increasing the $C_6H_5O_2$ + $NO_2$ rate, but peaking at slightly lower ambient $NO_x$ concentrations. Mechanism outcomes are only mildly sensitive to changes in the reactions of nitrophenols, a pathway implemented in GC13 and MECCA. Other novel aspects of GC13, including the 0% yield of organonitrates from the

5    reactions of NO with the initial bridged bicyclic peroxy radicals (following Xu et al. (2020)) and increased $HO_x$ recycling from the benzoylperoxy radical + $HO_2$ reaction, have minimal effects on ambient ozone and $HO_x$.



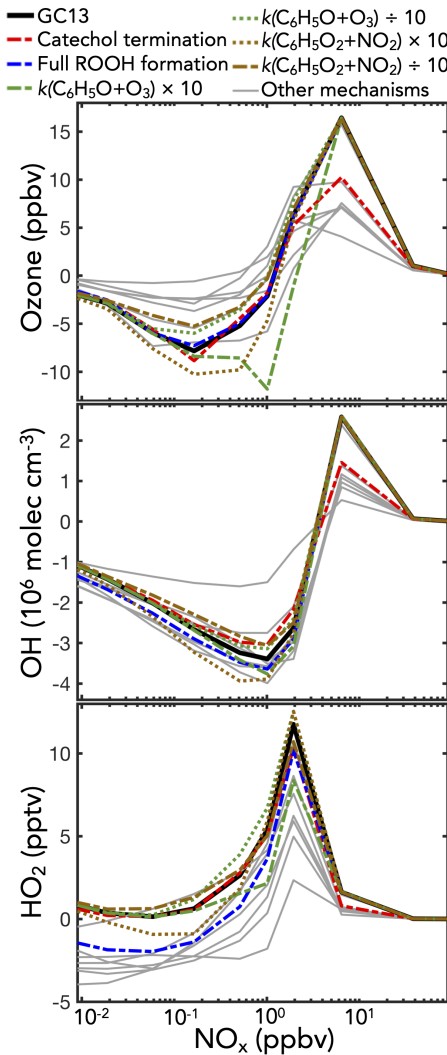

**Figure 9.** Sensitivity of midday ozone, OH, and HO$_2$ concentrations in the GC13 aromatic mechanism to changes in individual reaction rate constants and yields. The black lines show the standard GC13 box model simulation for the effect of mixed aromatic emissions in the continental boundary layer at 10:00-14:00 local time, relative to a simulation with no aromatic emission. The colored and patterned lines show the effects of individual changes in the GC13 mechanism. Gray lines show the results from the other mechanisms in Table 1. Results shown for the GC13 standard simulation and other mechanisms are the same as for Figure 8 except that BTX emissions were doubled to better show the changes from the individual reactions.



## 5  Global effects of aromatic oxidation

### 5.1  Methods

To investigate more deeply the effects of the BTX oxidation on atmospheric chemistry, we implement the GC13 mechanism into the GEOS-Chem CTM and compare it to selected other mechanisms in the GEOS-Chem environment. GEOS-Chem is
driven by meteorology from the Modern-Era Retrospective analysis for Research and Applications, Version 2 (MERRA-2) assimilation product of the NASA Global Modeling and Assimilation Ofice (GMAO). We use GEOS-Chem version 12.3 (DOI 10.5281/zenodo.2658178) with added $C_2H_4$ and $C_2H_2$ chemistry (Kwon et al., 2021) as a base, which includes 196 species in its chemical mechanism (not including aromatic chemistry), of which 149 are advected. We run global simulations at $2° \times 2.5°$ horizontal resolution with 47 vertical layers. For each simulation, we perform an initial 8-month spinup (1 Mar – 1 Dec 2015),
followed by one year of simulation from which seasonal and annual averages are output. We conduct one simulation with no aromatic emissions as a base case, and one simulation with GC13 for comparison. For simulations with GC13 chemistry, Henry's Law coefficients of newly included species (Table S1) are taken from Sander (2015) and Cabrera-Perez et al. (2016) for use in GEOS-Chem dry and wet deposition modules.

Anthropogenic VOC emissions in our GEOS-Chem simulations are from the Community Emissions Data System (CEDS)
Hoesly et al. (2018), overwritten with the Multi-resolution Emission Inventory for China (MEIC; Zheng et al. (2018)) and with the KORUS v5 inventory for the rest of East Asia (Woo et al., 2012; Jang et al., 2020). Biogenic emissions in GEOS-Chem are from the Model of Emissions of Gases and Aerosols from Nature (MEGAN) version 2.1 (Guenther et al., 2012), and open fire emissions are from the Global Fire Emissions Database (GFED) version 4 (van der Werf et al., 2010). Global annual emissions are 7.23 Tg, 10.42 Tg, and 7.30 Tg for benzene, toluene, and xylene from anthropogenic sources, and 1.67 Tg, 0.88
Tg, and 0.26 Tg respectively from biomass burning. Total BTX emissions are 60% higher than in the global model simulation of Taraborrelli et al. (2021), but only 4% higher (in carbon mass) than their total emissions of $C_6$-$C_9$ aromatics (including phenol, benzaldehyde, ethyl benzene, and lumped $C_9$ aromatics).

We also implement the two simplest alternative mechanisms, RACM2 and MOZART-T1, in the GEOS-Chem environment. MOZART-GC and SAPRC-11 were previously implemented in GEOS-Chem by Porter et al. (2017) and Yan et al. (2019)
respectively, but neither were incorporated into the standard version of GEOS-Chem. Instead, aromatic chemistry previously implemented in the standard version of GEOS-Chem was simply parameterized to achieve reasonable glyoxal and methylglyoxal yields with fixed branching ratios for SOA and peroxyacetylnitrate (PAN) formation (Fu et al., 2008; Fischer et al., 2014). Comparison of GC13 to this parameterized GEOS-Chem aromatic chemistry and to MOZART-GC is shown in the SI (Section S7-S8) for reference to past GEOS-Chem studies.

### 5.2  Effects on oxygenated VOCs

Figure 10 shows the impact of GC13 aromatic chemistry on concentrations of glyoxal, methylglyoxal, and formic acid in the lowest 1 km of the atmosphere. Aromatic oxidation increases the tropospheric production of these three oxygenated VOCs by 30%, 5%, and 9% respectively. Although absolute changes are strongest in source regions, the relative contribution of



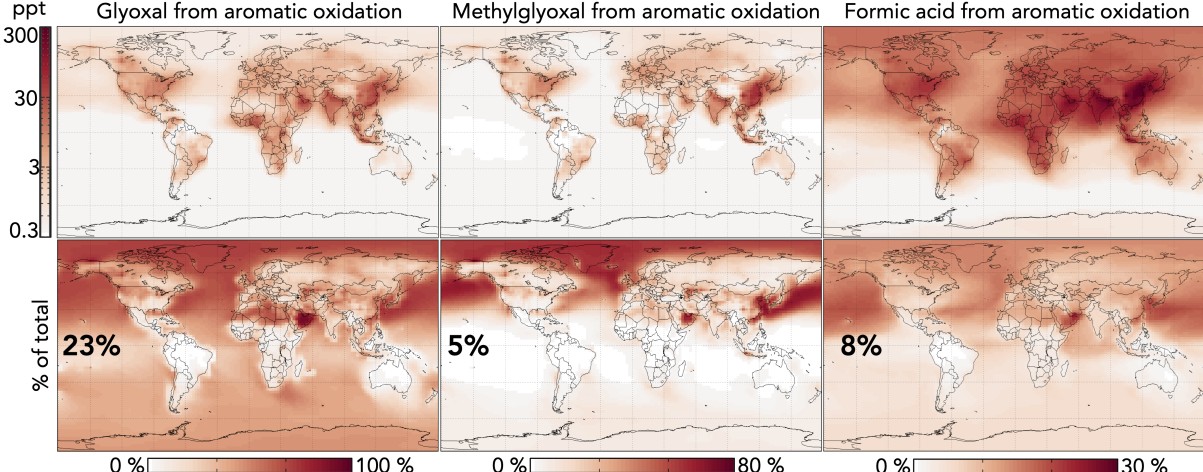

**Figure 10.** Effects of GC13 aromatic chemistry on glyoxal, methylglyoxal, and formic acid concentrations in GEOS-Chem. The top panels show absolute differences in annual mean concentrations below 1 km altitude between GEOS-Chem simulations with GC13 aromatic chemistry and with no aromatic chemistry. The bottom panels show the percentage contributions of aromatic chemistry to the total model concentrations of the three species. Color scales are logarithmic for top panels and linear for bottom panels.

aromatic chemistry to these concentrations extends globally, due both to later-generation production and to the longer lifetimes of aromatics relative to other precursors (e.g. anthropogenic alkenes and isoprene). Changes to gas-phase acetic acid are similar but smaller in magnitude to those of formic acid, with aromatic chemistry increasing production by 5%. Because these oxygenated VOCs are also formed in isoprene oxidation, the relative contribution of aromatics is much lower in high-isoprene

regions, especially the tropics. In the Middle East, aromatics are responsible for >80% of glyoxal, which may play a critical role in aqueous SOA formation. These findings for glyoxal and methylglyoxal are spatially consistent with Taraborrelli et al. (2021) but generally smaller, due in part to differences in the mechanisms (higher long-term yields of glyoxal and methylglyoxal from MECCA; see Figures S3-S4) and potentially to differences in non-aromatic glyoxal and methylglyoxal sources between the models. Neither Taraborrelli et al. (2021) nor Yan et al. (2019) discuss the contribution of aromatic chemistry to gas-phase

formic or acetic acid budgets.

      Additional effects from aromatic chemistry on the global distribution of oxygenated VOCs are shown in Figure 11. PAN is subject to competing influences: methylglyoxal formation from aromatic oxidation increases the source strength of the acylperoxy radical, the organic precursor to PAN, while lower $NO_2$ due to phenoxy-phenylperoxy cycling tends to decrease PAN production. The former effect dominates in source regions; aromatic oxidation with the GC13 mechanism increases PAN

mixing ratios over Northern Hemisphere continents by up to 40%. Downwind, particularly over oceans, PAN decreases due to phenoxy-phenylperoxy consumption of $NO_2$.

      Formaldehyde exhibits a similar spatial pattern to PAN, with competing effects from its direct secondary production via aromatic oxidation, leading to locally increased mixing ratios of up to 12% from aromatic oxidation, and indirect decreases due to reduced OH, which dominates downwind. Global formaldehyde production changes by just -0.1% from aromatic oxidation.





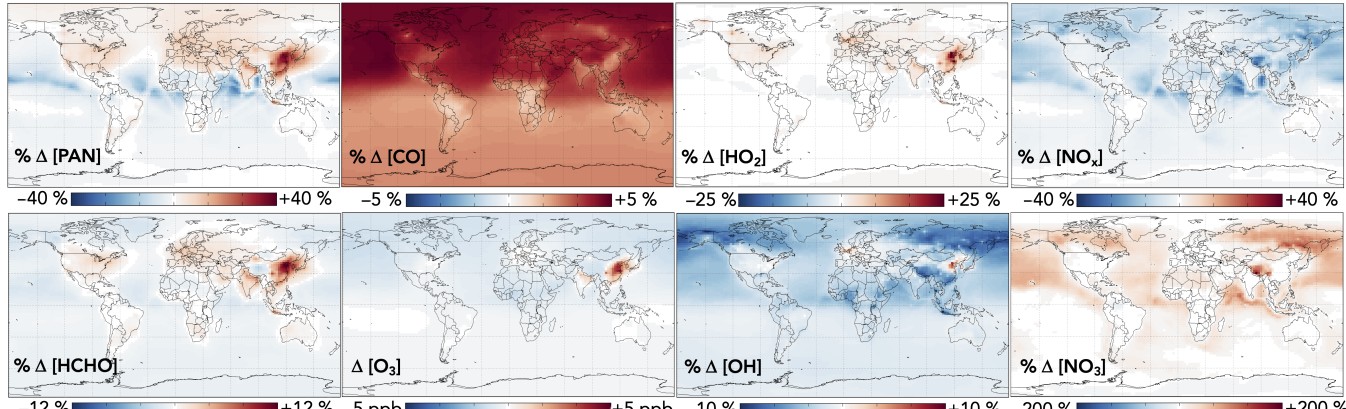

**Figure 11.** Effects of GC13 aromatic chemistry on PAN, formaldehyde, CO, ozone, HO₂, OH, NOₓ, and NO₃ concentrations. Panels show the absolute (for ozone) and relative (for others) differences in annual mean concentrations below 1 km altitude between GEOS-Chem simulations with GC13 aromatic chemistry and with no aromatic chemistry. Color scales are linear.

GC13 increases the tropospheric CO burden relative to the simulation without aromatic chemistry by 3%, due both to direct production and the decreased OH sink, with less spatial heterogeneity than other effects. The changes to both formaldehyde and CO are spatially consistent with the findings of Taraborrelli et al. (2021) using MECCA, but are smaller in magnitude, driven primarily by the smaller change in tropospheric OH in GC13.

Figure 12 compares the effects of aromatic chemistry on oxygenated VOC concentrations in GEOS-Chem with GC13 to simulations with the MOZART-T1 and RACM2 mechanisms. The most prominent difference between the simulations is the higher overall yield of glyoxal from aromatic oxidation in GC13, especially in later-generation chemistry. This results in increases of up to 60% in surface glyoxal when switching from either MOZART-T1 or RACM2 to GC13, with the strongest effects over the Middle East (where a lack of biogenic emissions renders aromatics the dominant glyoxal source) and in remote areas where later-generation chemistry dominates and decreases in OH increase VOC lifetimes. Overall tropospheric glyoxal loadings are 10% lower in MOZART-T1 and 13% lower in RACM2 than in GC13, while tropospheric glyoxal production from aromatics is 38% lower in MOZART-T1 and 61% lower in RACM2 than in GC13.

Methylglyoxal exhibits similar though less pronounced differences between the mechanisms, confined mostly to the northern hemisphere where its production is greatest. The strongest differences are seen for the MOZART-T1 mechanism, which produces only 62% as much methylglyoxal from BTX as GC13 in global simulations, resulting in decreases in surface methylglyoxal mixing ratio of up to 30% (2% globally). The RACM2 mechanism produces more methylglyoxal than MOZART-T1 (74% as much as GC13 globally), and therefore exhibits smaller changes (up to 10% decreases locally, 1% globally), and even some local increases. Differences in formaldehyde between the mechanisms are minor; global tropospheric formaldehyde is 0.4% higher with MOZART-T1 and 0.8% higher with RACM2 relative to GC13. The most prominent change is an increase of 5% in boundary layer formaldehyde over Northeast China with GC13 relative to MOZART-T1.



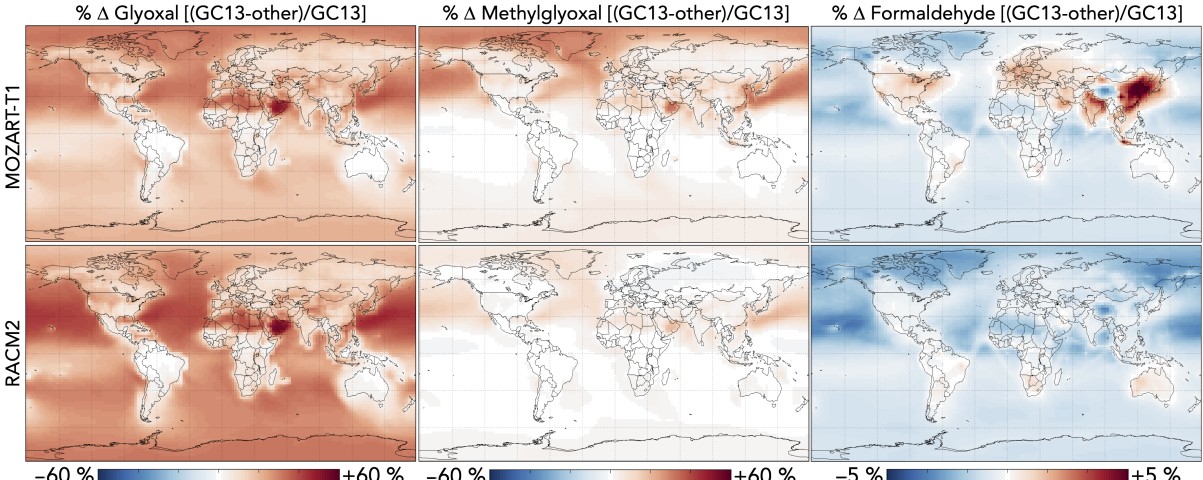

**Figure 12.** Differences in glyoxal, methylglyoxal, and formaldehyde concentrations between the GC13 mechanism and the MOZART-T1 and RACM2 mechanisms for aromatic chemistry. Values are percent differences in annual mean concentrations below 1 km altitude between GEOS-Chem simulations with the GC13 mechanism and with the MOZART-T1 or RACM2 mechanism. Color scales are linear.

### 5.3 Effects on oxidants

The effects of aromatic chemistry on radical and ozone budgets are shown in Figure 11. Impacts on $HO_x$ and ozone are consistent with the results of the continental boundary layer simulations in Section 4.3. $HO_2$ is increased by aromatic chemistry by up to 20% annually averaged in high-$NO_x$ aromatic source regions, but exhibits little change (<0.1%) globally. OH and
ozone both increase in high-$NO_x$ aromatic source regions, by up to 6% and 5 ppb respectively on annual averages, but decrease elsewhere, largely due to phenoxy-phenylperoxy radical cycling. On a global scale, these decreases slightly dominate; aromatic chemistry reduces tropospheric OH and ozone by 2.2% and <1% (0.37 ppb) on annual average. The effect on OH has a strong seasonal variation (Figure 13), with increases in source regions in the NH winter – up to 24% over Northeast China – but small effects and even slight decreases in the summer. This is due to the importance of carbonyl photolysis as a wintertime OH source
(Li et al., 2021).

NO$_x$ concentrations decrease everywhere as a result of aromatic chemistry, most notably in regions downwind of aromatic emissions in the Northern Hemisphere (Figure 11). Though PAN and peroxybenzoyl nitrate can act as NO$_x$ reservoirs, releasing NO$_x$ in remote air, there are additional NO$_x$ sinks from phenoxy-phenylperoxy cycling and nitrophenol formation. The impacts of NO$_3$ production from phenylperoxy + NO$_2$ are particularly pronounced in downwind regions with very low NO$_x$
concentrations, where this pathway can increase annual average NO$_3$ concentrations by up to 200%.

Generally, these effects of aromatic chemistry on oxidants in GC13 are consistent with those from MECCA in Taraborrelli et al. (2021), which also showed global decreases in NO$_x$, OH, and ozone from aromatic chemistry, along with local increases and seasonal cycles for OH and ozone in areas of strong aromatic emissions, and strong increases in NO$_3$. Global average changes tend to be stronger in Taraborrelli et al. (2021), especially for OH and ozone, consistent with the sharper decreases





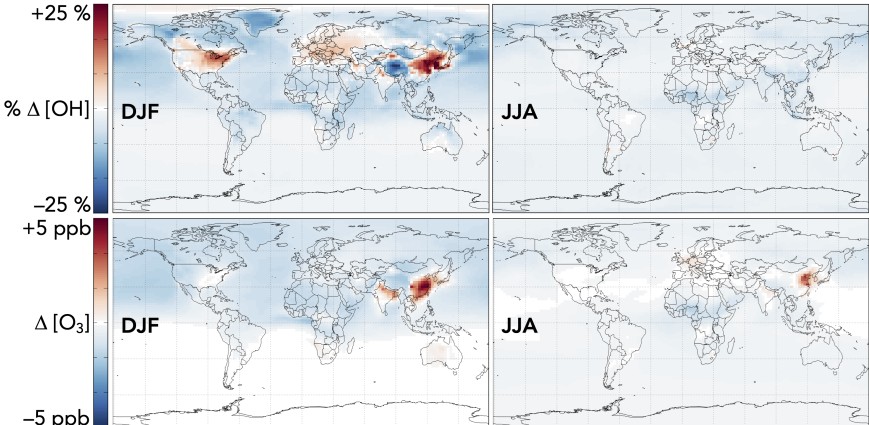

**Figure 13.** Seasonal effects of aromatic chemistry on OH and ozone concentrations. Panels show the relative (for OH) and absolute (for ozone) differences in mean concentrations below 1 km altitude between GEOS-Chem simulations with GC13 aromatic chemistry and with no aromatic chemistry for Dec. 1 2015 – Mar. 1 2016 (left) and Jun. 1 2016 – Sep. 1 2016 (right). Color scales are linear.

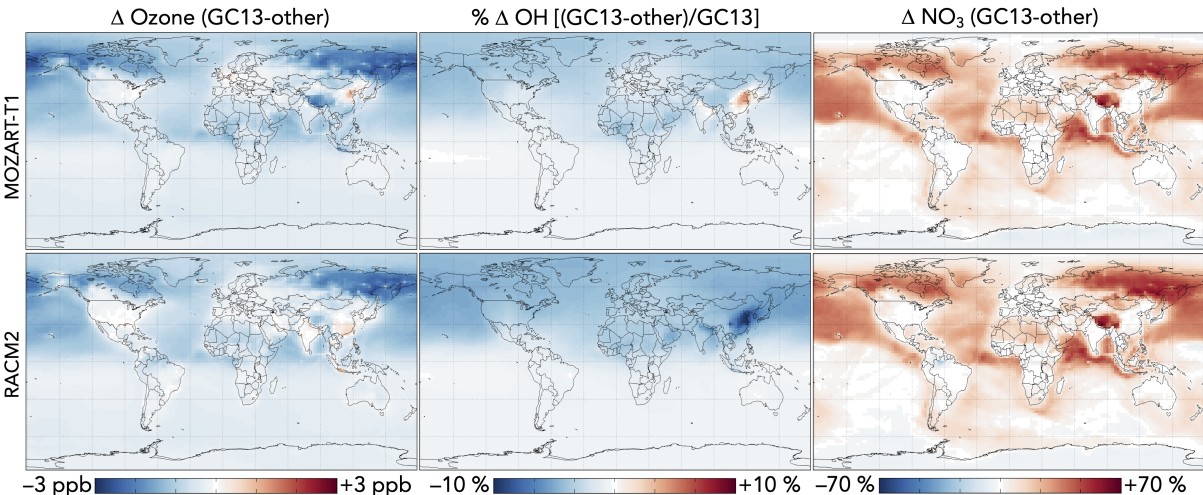

**Figure 14.** Differences in ozone, OH, and NO$_3$ concentrations between the GC13 mechanism and the MOZART-T1 and RACM2 mechanisms for aromatic chemistry. Values are absolute (for OH) or percent (for other species) differences in annual mean concentrations below 1 km altitude between GEOS-Chem simulations with the GC13 mechanism and with the MOZART-T1 or RACM2 mechanism. Color scales are linear.

in these compounds due to aromatic chemistry in continental boundary layer simulations (Figure 8). By contrast, SAPRC implemented in GEOS-Chem by Yan et al. (2019) showed increases in ozone, OH, and NO$_x$ concentrations as well as decreases in NO$_3$. We attribute this primarily to the absence of phenoxy-phenylperoxy cycling in SAPRC.



Differences in oxidant concentrations between global simulations with GC13 and with the MOZART-T1 and RACM2 mechanisms are shown in Figure 14. Changes to ozone, $HO_x$, and $NO_x$ between the mechanisms can be attributed primarily to the inclusion of phenoxy-phenylperoxy cycling and increased OH recycling in GC13. Surface ozone and OH are reduced in GC13 relative to MOZART-T1 and RACM2, consistent with findings in box model simulations (Section 4.3). Both MOZART-T1

and RACM2 cause increases in the tropospheric ozone burden relative to a simulation without aromatic chemistry, as Yan et al. (2019) also showed for SAPRC, while aromatic chemistry in GC13, like MECCA (Taraborrelli et al., 2021), causes a global ozone decrease. As an additional consequence of their lack of phenoxy-phenylperoxy cycling, the MOZART-T1 and RACM2 mechanisms simulate much lower tropospheric burdens of $NO_3$ (by 18% and 19% respectively) and slightly higher $NO_x$ burdens (by 3% and 2% respectively) than GC13. Finally, higher OH recycling in GC13 leads to local increases in OH

relative to MOZART-T1 (up to 4%) in source regions, but globally, differences in tropospheric and surface OH are <2%.

## 6   Conclusions

We developed a new compact mechanism (GC13) for fast and accurate simulation of benzene, toluene, and xylene oxidation chemistry in atmospheric models. Our mechanism includes only 17 unique species to describe the aromatic oxidation chain and 44 unique reactions. It includes recent information from experimental and computational studies, and captures the impor-

tant features of much more complex mechanisms. In particular, it incorporates recent evidence for efficient radical recycling from the bimolecular reactions of bridged bicyclic peroxy radicals, and explicitly treats phenoxy-phenylperoxy radical cycling as a sink for ozone and $NO_x$. Future development of a chemically detailed aerosol module could leverage the inclusion of methylcatechols and nitrophenols in GC13 as key SOA precursors.

We compared outcomes of GC13 to other aromatic oxidation mechanisms in box model simulations of environmental cham-

ber observations and of the continental boundary layer. Product yields from the mechanism exhibit good agreement with environmental chamber observations and result in increased glyoxal and methylglyoxal yields from aromatic oxidation relative to previous reduced mechanisms. Radical cycling in GC13 tends to increase simulated $HO_x$ radical concentrations, which past mechanisms have tended to underestimate (Bloss et al., 2005a; Carter and Heo, 2013). Phenoxy-phenylperoxy radical cycling decreases ozone production, which past mechanisms have tended to overestimate (Bloss et al., 2005a). We find that the

effects of aromatic chemistry on $HO_x$, $NO_x$, and ozone are strongly sensitive to uncertainties in the chemistry of the phenoxy-phenylperoxy system, with smaller but significant sensitivities to radical recycling from bridged bicyclic peroxy radicals and the oxidative fates of catechols.

We implemented the GC13 mechanism in the GEOS-Chem global atmospheric chemistry model and compared results to simulations without aromatic chemistry and with alternative mechanisms (MOZART-T1, RACM2). Aromatic oxidation plays

a particularly important role in the tropospheric budgets of small dicarbonyl species, contributing 23% and 5% to the global production of glyoxal and methylglyoxal, respectively. These values are substantially higher than those simulated with the MOZART-T1 and RACM2 mechanisms. Formaldehyde concentrations over Northeast China increase by 12% due to aromatic chemistry. Formic and acetic acids increase globally by 9% and 5%, respectively, through oxidation of ketene-enols generated





by aromatic ring-breaking. Aromatic chemistry decreases global tropospheric OH by 2.2% and ozone by less than 1% but increases them in polluted environments in winter, such as in Northeast China where wintertime OH increases by 24% and surface ozone increases by 5 ppb.

5  *Code and data availability.*  GEOS-Chem and box model output, as well as code files for replication, can be found in the Harvard dataverse repository (DOI 10.7910/DVN/0UQYOI)

*Author contributions.*  K. H. B. compiled the GC13 mechanism and designed and carried out the modeling described herein. Y. Y., P. D. I., K. L., and K. H. B. prepared the other mechanisms for implementation in KPP. K. H. B. prepared the manuscript with substantial assistance from D. J. J.

10  *Competing interests.*  The authors declare that they have no conflict of interest.

*Acknowledgements.*  This work was funded by the US EPA Science to Achieve Results (STAR) Program. K. H. B. acknowledges the support of the Harvard University Center for the Environment and the National Oceanic and Atmospheric Administration's Climate and Global Change Fellowship Programs.



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
