# Peer review of "Development and evaluation of a new compact mechanism for aromatic oxidation in atmospheric models"

_Atmospheric Chemistry and Physics, 2021_

## Author Comment (AC1)

We thank both the reviewers for their careful consideration of our manuscript and their helpful comments. We have reproduced the reviewer comments in full below, where we address each question and concern individually. Reviewer comments are written in black, our responses are in blue, and new text added to the manuscript is italicized.

REVIEWER 1:

Bates et al. have developed a new gas-phase chemical mechanism to simulate the oxidation chemistry for benzene, toluene, and xylenes in box and 3D models. They evaluated their model against chamber measurements as well as benchmarked it against other competing mechanisms. They find that their reduced mechanism performs well while noting that their mechanism alone models radical recycling from phenoxy-phenylperoxy that has modest effects on concentrations of smaller oxygenated products and oxidants. They also present interesting results from simulations performed separately with a boundary layer and 3D model.

Aromatics are an important class of organic compounds relevant for both anthropogenic (e.g., traffic) and biogenic (e.g., biomass burning) sources. Hence, the focus on aromatics and the development of a compact mechanism to simulate its atmospheric chemistry is well motivated. The manuscript was very easy to read and follow and I commend the authors for putting together a superb paper. I am less conversant with the chemical reactions described in the methods section so I am going to let the other reviewer(s) and editor to directly assess that section of the manuscript. However, I was able to follow the dominant chemical pathways and oxidation products and how they helped explain the results presented later. I recommend this paper for publication to ACP noting that the authors should consider my comments below before final submission.

1.        *Beyond BTX: My sense is that the impacts described here are likely to be biased (mostly underestimated) for the following reasons. First, how are aromatic species other than benzene, toluene, and xylenes accounted for in the modeling? My sense is that C8 and higher aromatics are lumped into xylenes. Or are they parsed by reactivity with OH? What about the treatment of multi-ring aromatics (e.g., naphthalene)? **This attribution (or lumping) of non-BTX species to BTX surrogates needs to be made clear throughout this work, in addition to how this attribution is justified and/or adds to the uncertainty.** Furthermore, what would be good to consider here would be to document the relative magnitude of non-BTX species to BTX. Also, are the authors concerned that certain species (e.g., furan in SAPRC) that are assigned to an aromatic surrogate (in the case of furan and SAPRC, to xylene) for computational reasons skew the model findings presented in this work? (It should not be a concern for the box modeling, correct?) Second, recent combustion emissions work has pointed out that gasoline exhaust contains a wide variety of single-ring aromatic hydrocarbons separate from BTX species (e.g, Zhao et al., ES&T, 2016; Drozd et al., ES&T, 2019; Lu et al., ACP, 2018). This seems to be less true for diesel exhaust. These non-BTX species have been shown to be very important for SOA formation and, in some models, have been accounted for as intermediate volatility organic compounds (IVOC). Third, biomass burning has been shown to be an important source of oxygenated aromatics including phenols and methoxyphenols (e.g., Koss et al, ACP, 2018; Hatch

et al., ACP, 2018). Since some of these biomass burning aromatics feature in the chemical mechanism presented in this work as intermediate oxidation products (e.g., phenol, catechol), I believe that they are likely to have similar effects as the explored BTX species. It's likely that the emissions of these oxygenated aromatics are considered in the GEOS-Chem inventories but it has not been made clear. All 3 factors described above will tend to generally increase (?) the effect size for aromatics.

First, to clarify, neither the box nor global model simulations lump larger aromatics or non-aromatics into the BTX chemistry described here. C9+ aromatics are simply not included in the chemistry of the box and global models, so their contributions are not artificially boosting the simulated importance of BTX. Multi-ring aromatics are only included in GEOS-Chem insofar as they contribute to SOA via emission of a naphthalene-like IVOC surrogate (see Pai et al., 2020, and Pye et al., 2010), and do not participate in GC13 gas-phase chemistry. As the reviewer notes, in some implementations of SAPRC furans are lumped with xylenes, but that is not the case here. Thus, we are not concerned that GC13 chemistry of non-BTX species affects the chemistry described in this manuscript. To clarify this, we have added the statement "*No emissions of other gas-phase aromatics are included*" to section 5.1

We agree, though, that the oxidation of larger aromatic species can be an important atmospheric contributor to OVOC budgets, oxidant and radical chemistry, and SOA production. Including the emissions and chemistry of these species (and emissions of oxygenated aromatics already in GC13) would likely, as the reviewer notes, increase the effect sizes shown in Figures 10-14. We initially considered such species beyond the scope of this manuscript, but based on the reviewer's interest have decided to add a small additional section (6) describing the potential importance of some larger aromatics and providing recommendations for their implementation in GEOS-Chem and other models with a heavily simplified mechanism:

" *Our GC13 mechanism and its implementation in GEOS-Chem focused on the effect of BTX emissions. Taraborrelli et al. (2021) estimated that non-BTX aromatics contribute 54% of global aromatic emissions by carbon mass with $C_6$-$C_8$ oxygenates, $C_8$-$C_9$ aromatic hydrocarbons, and higher aromatics each contributing approximately one thirds of this non-BTX fraction. The contributions from $C_6$-$C_8$ oxygenate emissions including benzaldehyde, phenol, cresols, and catechols can be readily accounted for in GC13 since they are already included explicitly as secondary species. $C_8$-$C_9$ aromatic hydrocarbons including ethylbenzene, styrene, and trimethylbenzenes can be added as independent species using simplified initial oxidation reactions to convert them directly to products already included in GC13. We have provided such reactions, simplified from MCM (Jenkin et al., 2003), in Section S1 of the SI, and this would add 3 species and 7 reactions to GC13. Higher aromatics such as polycyclic aromatic hydrocarbons (PAHs) are semi-volatile and their oxidation products partition heavily into the aerosol phase (Chan et al., 2009; Chen et al., 2016), so their simulation is more relevant to SOA formation than to oxidant chemistry.*"

2.      *Vapor wall losses: Page 10, line 4: Really? Recent work has argued that the timescales for vapor losses to the chamber walls are on the order of 5 to 15 minutes for a chamber volume

of ~10 m$^3$. Since these losses have been shown to be important for SOA formation (Zhang et al., PNAS, 2014; Cappa et al., ACP, 2016; Akherati et al., ACP, 2019), I would expect them to have a similar effect on gas-phase chemistry and radical and oxidant concentrations. Further, Zhang et al. (2014) show that the vapor wall loss effect on SOA formation is different under different NOx conditions. Perhaps this mechanism can shed light on why that might be? Vapor wall losses can be modeled relatively easily following the work of Zhang et al. (PNAS, 2014) and Krechmer et al. (ES&T, 2016).

Vapor wall losses can indeed be critically important for measured SOA yields, but GC13 does not seek to represent SOA formation nor to explicitly represent the wall-loss-prone compounds that contribute directly to SOA. While such semivolatile intermediates may be sources of OVOCs, even without wall losses their influence is likely to be small early in the chamber oxidation process, when most reactivity is attributable to the parent hydrocarbon. We purposely isolate early product yields from both simulations and experiments to avoid the longer-term, multi-generational chemistry that may be affected by the loss of intermediate products to the walls. Effects of semivolatile intermediates on oxidant concentrations and radical dynamics are also expected to be minor early in the experiment, and we do not use chamber simulations to constrain oxidant effects in the mechanism. Shedding light on the role of vapor wall losses in the NO$_x$-dependence of SOA yields is beyond the scope of this work and beyond the focus of the highly condensed GC13 mechanism; an improved representation of SOA from aromatics in GEOS-Chem is the subject of ongoing work independent of GC13. To clarify these points, we have expanded our discussion of vapor wall losses in section 4.1: "*Wall losses of semivolatile gases are not represented in these simulations, as we do not seek to model SOA formation or the yields of direct SOA precursors. An improved representation of SOA from aromatics in GEOS-Chem will be the subject of future work. While it is possible that wall losses of semivolatile intermediates affect experimental yields of OVOCs, this multigenerational chemistry is expected to play only a minor role at the short time scales isolated here.*"

3.      *Chamber simulation results and comparisons (Figures 4-6): While I understand the rationale behind the study design to simulate a representative chamber experiment, this work should also explore sensitivity in model predictions (limited to GC13 perhaps) to a broader range of input conditions observed across the chamber experiments used to evaluate the mechanism. For instance, are the model predictions sensitive to whether one assumes a constant OH profile chamber experiment (relevant for OH produced from photolysis of H2O2) or an OH profile that decays with time (relevant for OH produced from photolysis of HONO)? Are the model predictions sensitive to the chamber lights (e.g., intensity, spectrum) that should control the time-varying concentrations of NO and NO2? How do the model predictions change if the results were presented for a different time point (instead of 20 minutes after lights on)? How about the influence of chamber size (5 m3 vs. 30 m3) and mode of operation (batch mode vs. steady state model)? This mechanism sensitivity will be useful in interpreting the comparison with observations.

We were also curious about these various sensitivities and conducted a large number of such simulations to test our box model setup before arriving at the "default" simulations shown

in Figures 4-6. However, the focus of this work is on the mechanisms themselves (and sensitivities therein) rather than the parameters by which chamber experiments are conducted, so we did not opt to show these sensitivities in the main manuscript, but we did include results from many such sensitivity simulations in the SI. For the most part, initial yields reported in section 4.2 exhibit only minor sensitivities to the parameters we varied, which is why we did not elaborate on them in the main text. The most interesting sensitivities we found were to experiment duration (Figs. S3-5), temperature (Figs. S9-14), and initial VOC concentration (Figs. S15-S20), most of which were minor, although GC13 does exhibit some sensitivity of glyoxal and formic acid yields from benzene and of formaldehyde yields from toluene/xylene to initial VOC concentration. We have expanded this section of the SI by adding Figures S25-26 showing sensitivities to light intensity and oxidant concentration, and by providing greater descriptive detail in the accompanying text of Section S4. Furthermore, we have elaborated on these sensitivities in the main text and provided additional references to the SI: *"'Initial' yields shown in Figures 4-6 are after 10 min of photooxidation; additional results showing long-term yields after 24 h of oxidation are provided in Section S2 of the SI. Sensitivities to temperature and initial VOC concentrations are generally small, but additional results showing the effects of these parameters, as well as the effects of light intensity and oxidant source, can be found in the SI."* … *"We also find in GC13 simulations that the glyoxal yield from benzene is sensitive to the initial benzene concentration in chamber experiments (see Section S4 of the SI), which may further explain the spread of measured yields."* … *"long-term formaldehyde yields exhibit a temperature sensitivity of up to +3% K$^{-1}$"*. Finally, we have added a caveat to the beginning of Section 4.2 to drawn readers' attention

4.      \*Boundary layer and 3D model evaluation: The authors should discuss if and how the mechanism changes presented in this work have the potential to improve predictions of oxidants and gas-phase species that are likely to be biased based on past literature.

        We hope that a major benefit of GC13 and its documentation in this work will be its potential to improve such predictions in future applications, and we believe that the detailed descriptions of the mechanism, its sensitivities, and its comparison to previously used mechanisms already go some way to showing how GC13 can improve biases. We believe that extensive evaluation of the ambient box modeling and global simulations against atmospheric data are beyond the scope of this work, and a detailed assessment of past literature would be so specific to the conditions (*e.g.,* mechanisms and models used) of each study as to lose general interest. However, we have elaborated throughout the manuscript on points of comparison with previous studies, in hopes of drawing attention to the ways in which GC13 might change model outcomes, improve predictions, and alleviate biases. Added passages include:

        On ozone in box models: *"GC13 and MECCA have less ozone production than MCM, RACM, and SAPRC, all of which are known to overestimate peak ozone from chamber experiments (Bloss et al. 2005a, Goliff et al. 2013, Carter et al. 2013). … The reduced ozone formation in GC13 and MECCA relative to other mechanisms may improve model biases relative to chamber experiments (Bloss et al. 2005a, Goliff et al. 2013, Carter et al. 2013), and would*

*likely reduce the high simulated contribution of aromatics to ambient ozone formation in box model analyses of polluted environments (Luecken et al., 2018; Oak et al., 2019; Schroeder et al., 2020)."*

On $HO_x$ in box models: *"It has been reported that other mechanisms tend to underpredict $HO_x$ concentrations in simulations of chamber experiments (Chen, 2008), and both Bloss et al. (2005a) and Carter et al. (2013) comment on the need to increase HOx recycling from aromatic oxidation; GC13 thus brings $HO_x$ concentrations into improved alignment with chamber results."*

On global glyoxal: *"Our simulated contribution of aromatic oxidation to the budgets of glyoxal and methylglyoxal are substantially larger than in previous GEOS-Chem studies using simplified mechanisms with lower yields (Fu et al., 2008; Silva et al., 2018), but are more consistent with results from the detailed mechanisms in Taraborrelli et al. (2021) and Yan et al. (2019). The increased glyoxal yields in GC13 align with general model findings of negative biases relative to satellite observations of glyoxal columns in regions with strong anthropogenic influence (Chan Miller et al., 2016; Liu et al., 2012; Myriokefalitakis et al., 2008; Silva et al., 2018; Stavrakou et al., 2011)."*

5.      Page 10, line 30: I have always wondered what the NOx concentrations are in 'low NOx' chamber experiments. Literature from the Caltech group led by Prof. John Seinfeld has reported low NOx experiments to have NOx concentrations under 2 ppbv, reflecting the limits of quantification for their NOx analyzer. Assuming other chamber groups have encountered the same quantification problem with NOx, I wonder if a '10 pptv' assumption is justified here.

We agree with the reviewer that this can frequently be an issue in chamber experiments, and that assumptions of "$NO_x$-free" conditions may not always be verifiable with limited instrumentation. We have included error bars on reported $[NO_x]_0$ values where instrumental uncertainties are given, and have added a brief discussion of this point to the text in section 4.2: *"Experiments conducted with no added $NO_x$ are shown at $[NO_x]_0$ = 10 ppt (below which modeled yields are invariant with $[NO_x]_0$) regardless of whether $NO_x$ concentrations were monitored during the experiment. In some cases, $NO_x$ may off-gas from chamber walls (Carter et al., 2005). When reported, instrumental uncertainties on measured $NO_x$ mixing ratios are shown as horizonal error bars on experimental points."*

6.      Page 11, line 6: Here and elsewhere, I would encourage the authors to be quantitative on what 'low-to-moderate' NO/NOx and 'high' NO/NOx means.

We thank the reviewer for pointing this out; references to "high" and "low" NO/NOx can often be vague or have differing cutoffs depending on context. We have scoured the manuscript for mentions of high and low $NO_x$ and, where possible and appropriate, clarified by adding quantitative thresholds. The passage singled out by the reviewer has been amended to read: *"fixed yields of 53-57% fit most data under low-to-moderate NO conditions, but when *

*high* [NO$_x$]$_0$ *exceeds 100 ppb the observed yield declines*," and numerous other statements have been edited similarly (please see the track-changes document).

REVIEWER 2:

Bates et al. present work on the development of an updated chemical mechanism for aromatic chemistry for use in computational models of atmospheric chemistry. They demonstrate the mechanism performs well as assessed against state-of-the-art laboratory measurements and compare the simulated abundances of compounds like glyoxal, OH, and ozone across different implementations of aromatic chemistry. This work is scientifically sound, and the manuscript is generally well written. Barring one major accessibility issue, I recommend that this paper is accepted for publication following a few minor comments below.

**Colour blindness and Figures 4-9**: Figures 4-9 are not intelligible to colour blind readers. I strongly suggest the authors remake these figures to make them more accessible. There are a range of tools available online to explore if a given colour palette or figure is "colour blind friendly".

We thank the reviewer for bringing this important point to our attention. Figures 4-9 have been re-rendered with a more accessible Matlab color palette that is distinguishable to readers with deuteranopia, protanopia, or tritanopia (http://mkweb.bcgsc.ca/biovis2012/color-blindness-palette.png).

**Computational Impact**: A major component of this work relates to the development and justification of the simplified mechanism, which is well justified in the manuscript given the described issues associated with computational limitations. It is thus surprising that the authors do not quantify the impact of the mechanism on global model simulation times. While this work likely improves the representation of chemistry in GEOS-Chem, what is the computational cost associated with the increased transport and chemistry demands? If I were a model developer, what are the costs of using this mechanism?

We agree with the reviewer that this is an important point to address, but it can be difficult to quantify in a meaningful way. The computational cost of the added aromatic chemistry will vary between users with a variety of simulation parameters (*e.g.,* duration, spatial and temporal resolutions, mixing schemes, emission fields, and other customizable settings) and hardware parameters (since the CPU time of various components of the model scale differently with the number of cores used). However, we have added a statement in section 5.1 describing the computational cost of the aromatic chemistry: "*The added aromatic chemistry in the GC13 simulation increases overall CPU time by an average of 1.7% relative to the base simulation, attributable predominantly to gas-phase chemistry (64%) and transport (26%).*" We have also added a similar statement to the conclusions: "*GC13 increases the computational cost of an annual simulation by 1.7% relative to a simulation without aromatic chemistry.*"

**P20 L5:** Why are these results different from previous glyoxal simulations in GEOS-Chem (Silva et al 2018), which do not show a large change in glyoxal from aromatic oxidation over the Middle East?

Multiple factors may contribute to the different findings between Silva *et al.* (2018) and this manuscript. First, GC13 produces more glyoxal from aromatic oxidation than the previously implemented GEOS-Chem mechanism; this can be seen in greater detail in Figure S31. Second, the figures in Silva *et al.* (2018) show contributions of aromatics and other precursors to total column burdens of glyoxal, while this manuscript focuses on surface glyoxal. The short lifetimes of toluene and xylene confine their influence closer to the surface, while $C_2H_2$, which is simulated by Silva *et al.* (2018) to be the dominant contributor to total column glyoxal over much of the Middle East, has a longer lifetime and thus exerts more influence over free tropospheric glyoxal burdens. Third, the two studies use different anthropogenic emission inventories (EDGAR/RETRO with regional modifications in Silva *et al.*, CEDS here) in different years (2005 in Silva *et al.*, 2015-16 here), which changes the distribution of aromatic emissions as well as those of other glyoxal precursors. Finally, changes to oxidation mechanisms, SOA uptake, and other model mechanics between the GEOS-Chem versions used (11-02 in Silva *et al.*, 12.3 here) may affect simulated glyoxal. We have opted not to complicate the manuscript with a detailed analysis of these differences, but have added a reference to Silva *et al.* to the discussion of simulated glyoxal from the implementation of GC13 in GEOS-Chem: "*Our simulated contribution of aromatic oxidation to the budgets of glyoxal and methylglyoxal are substantially larger than in previous GEOS-Chem studies using simplified mechanisms with lower yields (Fu et al., 2008; Silva et al., 2018), but are more consistent with results from the detailed mechanisms in Taraborrelli et al. (2021) and Yan et al. (2019)*".

**P9 L27 & P10 L5:** "Environmental chamber simulations" and "Continental boundary layer simulations" seem to be subtitles but aren't formatted as such.

Rather than format these as headers of sub-sub-sections, we have chosen to keep them as undifferentiated paragraphs within sub-section 4.1, much like the paragraphs describing each mechanism in section 3.

**P10 L30:** $[NO_x]_0$ is not defined in the manuscript.

We have added a definition in the first paragraph of Section 4.2: "...we present our results as a function of *initial* $NO_x$ *mixing ratios* ($[NO_x]_0$)."

**P18 L7:** Does Kwon et al. 2021 differ from earlier implementations of $C_2H_x$ chemistry into GEOS-Chem (e.g. Safieddine et al. 2017)?

The implementation of $C_2H_x$ chemistry in GEOS-Chem by Safieddine et al. (2017) was drawn from the mechanism in CAM-Chem as detailed in Lamarque *et al.* (2012). The $C_2H_2$ chemistry included in Kwon *et al.* (2021) and here is identical to that in Lamarque *et al.* (2012), while our $C_2H_4$ chemistry is slightly more complex (with the addition of a hydroxynitrate and an

alkoxy radical), following MCM (Saunders *et al.,* 2003) and IUPAC (Atkinson *et al.,* 2004) recommendations. Because the mechanisms are provided in detail in Kwon *et al.* (2021), we choose not to describe them further here.

[revised manuscript text omitted]